# DynST: Large-Scale Spatial-Temporal Dataset for Transferable Traffic Forecasting with Dynamic Road Networks

## Abstract

In real-world traffic networks, it is common to encounter a shortage of historical data in the target region. Researchers often address this issue through transfer learning. However, transfer learning tasks in traffic prediction currently lack dedicated datasets and instead rely on datasets designed for non-transfer prediction tasks. The major drawback of these existing datasets is the adoption of a fixed network topology to model the real world's road networks. This does not align with reality and limits the model's transferability. To tackle this issue, we propose DynST, a dataset specifically designed for transfer learning tasks in traffic prediction, with a massive data volume of 20.35 billion, spanning 20 years and 9 regions. The key feature of DynST is evolving dynamic road network topology, which reflects the evolution of real road networks. Moreover, to address the shortcomings of the distance-based adjacency generation algorithm, we introduce a novel tree-based algorithm. Extensive experiments demonstrate that the adoption of DynST as the source dataset can significantly enhance the performance of the target region. The comparative experiment also validates that our adjacency matrix generation algorithm can lead to improved prediction accuracy. We believe that DynST, with rich spatial variation information, will facilitate research in the field of transfer traffic prediction.

## 1 Introduction

Traffic forecasting plays a crucial role in urban management and travel planning, and relies heavily on historical data. In reality, the available data in the target areas are often inadequate. A feasible approach is to utilize the transfer learning technique, where generalizable knowledge learned from a data-rich region is transferred to the target area. Although many transfer learning models have been developed (Wang et al., 2021; Yin et al., 2022; Jin et al., 2022; Yao et al., 2019; Jin et al., 2022), there is currently no dataset specifically designed for the transfer learning task. These models typically leverage datasets proposed for non-transfer learning, which primarily suffer from the use of an invariant topological structure to describe the real-world road network (Yu et al., 2018; Li et al., 2017; Cui et al., 2019; Guo et al., 2019; Song et al., 2020; Liu et al., 2023b).

The existing datasets inherently assume the completeness of data and the invariance of the road network. In the real world, due to factors like weather conditions and equipment failures, data integrity is infeasible in large-scale datasets (Yu et al., 2024). Additionally, it is normal for regional road construction to evolve as the dataset spans long periods (Chen et al., 2001). On the other hand, the nature of transfer learning tasks requires knowledge to be transferred between two regions with completely different road networks, but not the same network topology. Intuitively, increasing the dynamism of the original region's road network topology can enhance model generalizability. By exposing the model to a more diverse topology of road networks, it can learn more robust patterns that are less tied to specific nodes' characteristics. In contrast, a fixed road network topology in the source region may limit the model's transferability to adapt to new network structures.

Motivated by this, we propose DynST, a large-scale spatial-temporal dataset featuring evolving dynamic topology with data spanning up to 20 years, specifically designed for transfer learning tasks in traffic forecasting. The basic information is compared in Table 1. To construct DynST, we sourced

Table 1: Comparison between DynST and existing datasets. **Degree**: the average degree of each node. **Meta**: the number of metadata files for each dataset. **Frames**: the total time frames. **Data Points**: multiplication of nodes and time frames. **M**: million ($10^6$). **B**: billion ($10^9$). We summarize the total number in the last row as DynST.

| Dataset | Nodes | Edges | Degree | Meta | Time Range | Frames | Data Points |
|---|---|---|---|---|---|---|---|
| PeMSD7(M) (Yu et al., 2018) | 228 | 1664 | 7.298 | 6 | 05/10/2012 – 06/30/2012 | 12672 | 2.89M |
| PeMSD7(L) (Yu et al., 2018) | 1026 | 14535 | 14.167 | 0 | 05/01/2012 – 06/30/2012 | 12672 | 13.00M |
| METR-LA (Li et al., 2017) | 207 | 1515 | 7.319 | 3 | 03/01/2012 – 06/12/2012 | 34272 | 7.09M |
| PEMS-BAY (Li et al., 2017) | 325 | 2369 | 7.289 | 3 | 01/01/2017 – 06/30/2017 | 52116 | 16.94M |
| Seattle Cui et al. (2019) | 323 | 1001 | 3.099 | 0 | 01/01/2015 – 12/31/2015 | 105120 | 33.95M |
| PEMS04 Guo et al. (2019) | 307 | 338 | 1.101 | 0 | 01/01/2018 – 02/28/2018 | 16992 | 5.22M |
| PEMS08 (Guo et al., 2019) | 170 | 276 | 1.624 | 0 | 07/01/2016 – 08/31/2016 | 17856 | 3.04M |
| PEMS03 (Song et al., 2020) | 358 | 546 | 1.525 | 1 | 09/01/2018 – 11/30/2018 | 26208 | 9.38M |
| PEMS07 Song et al. (2020) | 883 | 865 | 0.980 | 0 | 05/01/2017 – 08/06/2017 | 28224 | 24.92M |
| LargeST (Liu et al., 2023b) | 8600 | 201363 | 23.414 | 9 | 01/01/2017 – 12/31/2021 | 525888 | 4.52B |
| D03 | 1124 | 1129 | 1.004 | 575 | 03/09/2001 – 04/11/2024 | 2.41M | 1.46B |
| D04 | 2474 | 2497 | 1.009 | 202 | 12/04/2001 – 04/11/2024 | 2.28M | 4.26B |
| D05 | 427 | 426 | 0.998 | 63 | 03/25/2005 – 04/10/2024 | 1.99M | 0.35B |
| D06 | 614 | 614 | 1.000 | 104 | 09/03/2005 – 04/10/2024 | 1.94M | 0.61B |
| D07 | 2781 | 2811 | 1.011 | 89 | 01/01/2001 – 04/10/2024 | 2.33M | 5.54B |
| D08 | 1523 | 1533 | 1.007 | 1059 | 03/02/2001 – 04/10/2024 | 2.41M | 2.43B |
| D10 | 751 | 753 | 1.003 | 540 | 07/01/2006 – 04/10/2024 | 1.86M | 0.86B |
| D11 | 830 | 844 | 1.017 | 215 | 01/05/2001 – 04/03/2024 | 2.26M | 1.53B |
| D12 | 1696 | 1709 | 1.008 | 142 | 01/01/2002 – 04/10/2024 | 2.33M | 3.33B |
| **DynST** | **12220** | **12316** | **1.008** | **2989** | **01/01/2001 – 04/11/2024** | **19.81M** | **20.35B** |

data from the California Department of Transportation Performance Measurement System (PEMS) [1], aggregating 30-second granularity data to the standard 5-minute granularity, to align with prevailing practices. The dataset includes traffic flow, vehicle speed, and road occupancy information. The construction of DynST ensures the realities of real-world traffic data. Two primary considerations are addressed: (1) Road Network Development and Dynamic Missing Data: The PEMS system irregularly adds new sensors to accommodate regional road development. Due to various unexpected events, missing data in the deployed sensors are inevitable. (2) Data Cleaning for Reliability: The cleaning process is supposed to ensure data reliability and maintain a sufficient volume for training purposes. We filter out any sensors with missing values throughout the day. Subsequently, we use the latitude and longitude coordinates of the sensors, along with an adjacency matrix generation algorithm, to create the road network topology. In other words, DynST ensures that the sensors and the road network structure maintain consistency throughout the day, while also exhibiting some variation from one day to the next. This strategy cleverly constructs a large-scale spatial-temporal dataset with a dynamic road network structure. It is more reflective of reality while also capturing regional road changes and various emergencies.

During the data construction, annotating the adjacency matrix is a crucial step. Manually annotating large datasets is impractical, and the distance-based generation method proposed by Liu et al. (2023b) has overconnected and disconnected issues. To obtain an adjacency matrix that closely reflects the actual road network, we propose a tree-based generation method to address the shortcomings.

Our research adheres to PEMS regulations [2], and the dataset is provided under the CC BY-NC 4.0 License. Our contributions can be summarized as follows: (1) We constructed a large-scale dataset, DynST, specifically designed for transfer learning tasks in traffic forecasting, which features a dynamic road network structure; (2) We proposed a tree-based adjacency matrix generation algorithm that creates a topology more closely aligned with real-world road networks; (3) Extensive experiments demonstrate that utilizing DynST can effectively enhance the predictive capability of transfer learning models in target regions. Comparative experiments validate the effectiveness of the proposed adjacency matrix generation algorithm.

---

[1] https://pems.dot.ca.gov/
[2] https://pems.dot.ca.gov/?directory=Help&dnode=Help&content=var_terms

## 2 RELATED WORK

### 2.1 NON-TRANSFER LEARNING TRAFFIC FORECASTING

In the early days traffic forecasting tasks were viewed as multivariate time series forecasting tasks and were modeled using statistical models like ARIMA (Williams & Hoel, 2003) and VAR (Lu et al., 2016). They could not handle complex spatial-temporal correlation. In recent years, the development of Spatial-temporal Graph Neural Networks (STGNNs) (Wu et al., 2020) has led to significant progress in traffic prediction tasks. STGNNs use GNNs (Kipf & Welling, 2017) to model spatial information, and a variety of methods adopt sequential models to model the temporal information, such as RNN-based methods (Li et al., 2017; Bai et al., 2020; Pan et al., 2019), CNN-based methods (Yu et al., 2018; Wu et al., 2019; Song et al., 2020; Han et al., 2021), attention-based methods (Zheng et al., 2020; Cirstea et al., 2022; Jiang et al., 2023; Guo et al., 2019; 2022; Li et al., 2022; Liang et al., 2018). The recent research hotspot lies in the utilization of Transformer structure (Vaswani et al., 2017) to model spatial-temporal dependency, and the Transformer-based methods achieve superior performance (Guo et al., 2023; Jiang et al., 2023; Liu et al., 2023a; Xu et al., 2020). Beyond the aforementioned categories, there are also some methods (Fang et al., 2021; Choi et al., 2022) focused on graph representation construction. These methods have facilitated the development of traffic prediction tasks.

These above models require a lot of historical data to be trained from scratch, but unfortunately, collecting comprehensive traffic data in a particular city is a time-consuming and expensive endeavor. Therefore, transfer learning techniques become a promising direction to address this issue.

### 2.2 TRANSFER LEARNING IN TRAFFIC FORECASTING

Transfer learning is the process of enhancing performance in a target domain by learning rich generalized information from the source domain. This technology has been widely used in the field of computer vision (Dosovitskiy et al., 2021) and natural language processing (Devlin et al., 2019; Brown et al., 2020). The methods based on transfer learning in traffic prediction mainly adopt meta-learning (Yao et al., 2019; Jin et al., 2022) and pretrained-finetuning (Wang et al., 2021; Yin et al., 2022; Jin et al., 2022) strategies.

However, current transfer learning models do not have dedicated datasets; instead, they rely on datasets designed for non-transfer learning tasks. These datasets model urban road networks using fixed topology and do not explicitly provide the capability for the model to make predictions on changing road networks. However, the task inherently requires the models to transfer across different road networks. Therefore, we propose DynST, which adopts a novel evolutionary dynamic topology structure to describe real-world road networks.

## 3 PRELIMINARY

**Dynamic Road Network Structure.** The objective dynamic urban road network is modeled as a graph family, whose entry in day $d$ is denoted as $\mathcal{G}_d = (\mathcal{V}_d, \mathcal{E}_d)$, where $\mathcal{V}_d$ is the set of $N_d$ nodes denoting sensors after filtered out, $\mathcal{E}_d \in (\mathcal{V}_d \times \mathcal{V}_d)$ is the set of edges representing the connections between nodes, from which we derive the adjacency matrix $\mathcal{A}_d$ by proposed tree-based generation algorithm. Specifically, the existing datasets adopt a fixed topological structure to model the objective road network, i.e., $\forall d_i, d_j \in D, \mathcal{G}_{d_i} = \mathcal{G}_{d_j}$, where $D$ is the total temporal span of the dataset.

**Traffic Forecasting**. $\mathbf{X}_t \in \mathbb{R}^{N \times C}$ is the feature matrix at time step $t$, where $C$ is the feature channel, such as traffic flow, speed, and occupancy. The goal of traffic forecasting is to predict the future traffic signals based on the historical sequence, formulated as

$$[\mathbf{X}_1, \mathbf{X}_2, \ldots, \mathbf{X}_T] \xrightarrow{\mathbb{F}(\cdot|\Theta)} [\hat{\mathbf{X}}_{T+1}, \hat{\mathbf{X}}_{T+2}, \ldots, \hat{\mathbf{X}}_{T+T'}] \tag{1}$$

where $T$ represents the historical time span, $T'$ is the future time span and $\Theta$ is the model parameters.

**Transfer Learning Task**. For the transfer learning task in traffic forecasting, the road networks of the source and target region differ. Given the source region $[\mathcal{G}_1^{source}, \mathcal{G}_2^{source}, \ldots, \mathcal{G}_D^{source}]$ and target region $\mathcal{G}^{target}$, denote the corresponding feature matrix as $\mathbf{X}^{source}$ and $\mathbf{X}^{target}$. The goal of transfer learning is to improve the performance in $\mathbf{X}^{target}$ by leveraging the knowledge from $\mathbf{X}^{source}$.

# 4 DYNST

## 4.1 LIMITATIONS OF EXISTING DATASETS

**Fixed Road Network Structure**. As far as we know, in traffic prediction, there is no dedicated dataset specifically designed for transfer learning; instead, transfer learning models treat the naive datasets as the source region dataset. Transfer learning requires the model to make predictions across different road networks, but these datasets adopt a fixed topological structure to portray the road network, which cannot explicitly provide the model with transferability. In other words, since the road network structure in the source region dataset does not change, the model tends to overly rely on certain features, which restrain its transferability. Therefore, we posit that the adoption of fixed road network structure in existing datasets hinders models' generalizability.

**Distance-based Adjacency Matrix Generation Method**. Current datasets use the adjacency matrix to represent the road network topological structure, which is the root for a bunch of prediction methods (Wu et al., 2019; Bai et al., 2020; Yu et al., 2018; Guo et al., 2019). Recent studies, such as Liu et al. (2023b), have explored automated methods for generating adjacency matrices in traffic forecasting. One common approach is the distance-based method (Liu et al., 2023b), which assumes that the nodes are neighborhoods within a hard threshold. However, it often happens that two nodes within a close distance are not connected, i.e., there is no straightforward path to reach each other; or two nodes are adjacent but the distance coincidentally exceeds the threshold, and they would be determined as isolated nodes. So this distance-based adjacency matrix generation method would lead the model to learn the wrong topological relationship, which can be summarized as overconnected and disconnected issues.

## 4.2 DATASET CONSTRUCTION

We obtain the raw data from the PEMS system (Chen et al., 2001). The PEMS system reports raw data every 30 seconds. We aggregate it into 5-minute granularity, which means 12 data points an hour and 288 data points a day. We collected and cleaned out 9 districts' data. The dataset consists of three types of traffic signals: traffic flow, vehicle speed, and occupancy.

- Traffic flow is quantified as the number of vehicles detected by a sensor over five minutes.
- Vehicle speed is measured by the sensors as the average speed of vehicles during five minutes expressed in miles per hour (mph).
- Occupancy refers to the percentage of time in which vehicles are passing over the sensors.

**Metadata**. We provide detailed metadata for each district, which includes 12 attributions, such as the Sensor ID, lane information, latitude, and longitude. The latitude and longitude data can be used to construct the adjacency matrix. This metadata is presented in a timeline format and its update period is aligned with the new road construction. Additional detailed information is available in Appendix D. We hope rich metadata can facilitate future research with finer granularity.

**Data Cleaning**. After collecting the daily raw data via PEMS, we processed it for each day. Given abundant missing values (bigger than 50% in some cases) and the limitations of simple linear interpolation, which could distort the patterns, we grouped the raw data by Sensor ID and filtered out any data points with missing values in any of the three types of traffic signals. This strategy ensures the data accurately reflects real-world distributions. See more in Appendix A.

## 4.3 ADJACENCY MATRIX CONSTRUCTION

As mentioned in Section 4.1, the conventional distance-based adjacency matrix generation method determines the connectivity between two nodes by a predefined threshold. As shown in Figure 1, in areas with densely deployed sensors, the proximity of adjacent sensors frequently leads to overconnectivity and a redundant road network structure. Conversely, in zones where sensor deployment is sparse, the inability to form edges due to inter-sensor distances exceeding the threshold can result in isolated nodes and disconnected graph structure. Moreover, through prior knowledge obtained from PEMS system documentation and visualization analysis in Figure 5, the PEMS system collects traffic signals on expressways and highways. There are mostly straight roads (single chain graphs)

**Algorithm 1:** Adjacency Matrix Generation

**Input** : Vertex set $\mathcal{V}_d$, distance matrix $\mathcal{D}_d$, thresholds $\theta_1, \theta_2$, coefficient $k\,(0 < k < 1)$

**Output :** $\mathcal{G}_d = (\mathcal{V}_d, \mathcal{E}_d)$

1 **Function** minimumSpanningTree($\mathcal{V}'_d$, $\mathcal{E}'_d$):

2     **return** the minimum spanning tree generated from $\mathcal{V}'_d$ and $\mathcal{E}'_d$

3 **Function** shortestPathLength($\mathcal{G}_d$, $u$, $v$):

4     **return** the shortest path length from $u$ to $v$

5 $\mathcal{G}_d \leftarrow (\mathcal{V}_d, \emptyset)$

6 $\mathcal{E}'_d \leftarrow \{(u, v, \mathcal{D}_d[u][v]) | u, v \in \mathcal{V}_d, \mathcal{D}_d[u][v] < \theta_1\}$

7 $\mathcal{G}_d \leftarrow$ minimumSpanningTree($\mathcal{V}_d, \mathcal{E}'_d$)

8 $\mathcal{E}''_d \leftarrow \emptyset$

9 **foreach** vertex $u \in \mathcal{V}_d$, $\deg(u) \leq 1$ **do**

10     **foreach** vertex $v \in \mathcal{V}$, $u \neq v$ **do**

11         **if** $\mathcal{D}_d[u][v] < \theta_2$ **then**

12             $\mathcal{E}''_d = \mathcal{E}''_d \cup \{(u, v, \mathcal{D}_d[u][v])\}$

13 Sort edges in $\mathcal{E}''_d$ by increasing weight

14 **foreach** edge $(u, v, w) \in \mathcal{E}''_d$ **do**

15     **if** $u$ and $v$ are not connected in $\mathcal{G}_d$ **or** $\mathcal{D}_d[u][v] < k \times$ shortestPathLength($\mathcal{G}_d$, $u$, $v$) **then**

16         $\mathcal{E}_d \leftarrow \mathcal{E}_d \cup \{(u, v, \mathcal{D}_d[u][v])\}$

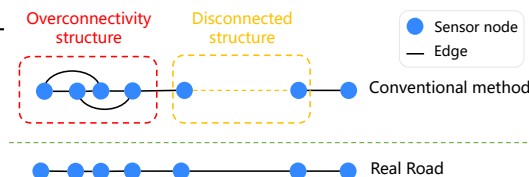

Figure 1: Conventional distance-based adjacency generation method leads to over-connected and disconnected issues.

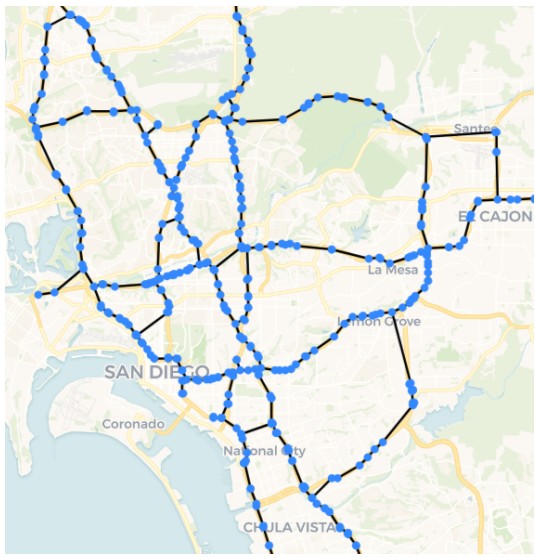

Figure 2: Generated road network topology visualization.

supplemented by a few crossings. Thereby, the degree of the nodes is close to 1.0. See more analysis about the assumption for the degree of a node in Appendix E.

To address this limitation and follow the principles, we propose a tree-based adjacency matrix generation algorithm, as its main part is described in Algorithm 1 and the rest are detailed in Appendix E. The core idea is to denote the distances between nodes (sensors) as edge weights to generate a minimum spanning tree (MST) in lines 1-2, followed by complementing the network structure based on shortest paths in line lines 3-4. Specifically, it initially filters out edges with distance weights bigger than 50 kilometers (lines 5-6). This threshold reduces the computational load while avoiding extreme connectivity. Then, it constructs an MST using these edges (line 7). Subsequently, to complete the network structure, for all isolated vertexes and pendant vertexes whose degree < 1, it identifies vertices within the threshold distance and adds their connecting edges to the set $\mathcal{E}''_d$ (lines 8-12). Edges in $\mathcal{E}''_d$ are then sorted by weight ascending; for each pair of vertices, $u$ and $v$, if they are not connected or if the distance between them is less than $k$ times the shortest path length between them in graph $\mathcal{G}_d$, the edge $u$–$v$ is incorporated into $\mathcal{G}_d$ (lines 13-16). So far, we have obtained the adjacency matrix for one day, and by repeating the process, we can obtain the dynamic road network topology for the whole dataset. Figure 2 visualizes the adjacency matrix, where blue nodes represent sensor locations and black lines represent edges. The results demonstrate that the adjacency matrix generated by our algorithm perfectly matches the actual road network structure.

### 4.4 DATA STATISTICAL ANALYSIS

We conducted a statistical analysis over the temporal and spatial dimensions, aiming to help researchers better understand the characteristics of DynST.

**Temporal Characteristics**. We draw a daily traffic signal distribution for each district in Figure 3. Traffic flow exhibits twin pronounced peaks on weekdays according to the morning and evening rush

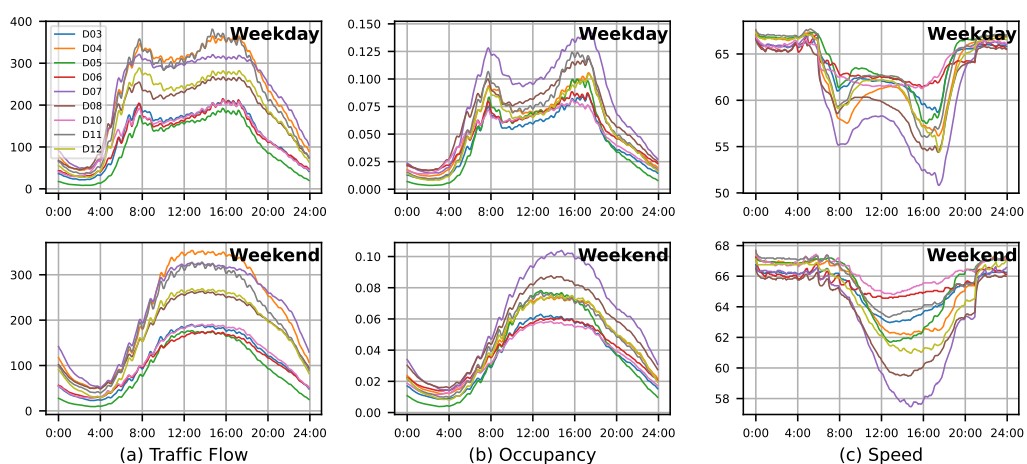

Figure 3: Illustration of weekly patterns of traffic signal distribution.

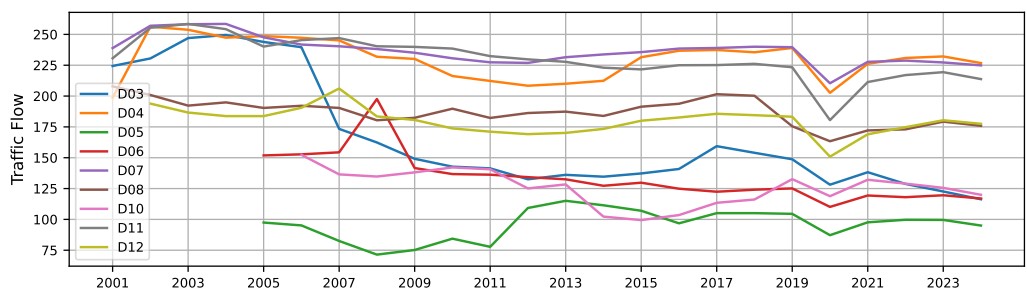

Figure 4: Annual trend of average traffic flow.

hours, and a sinusoidal-like curve on weekends (Figure 3 (a)). The occupancy (Figure 3 (b)) and speed (Figure 3 (c)) exhibit similar patterns to the traffic flow grouped by day of the week. Moreover, trends of occupancy and speed show an anti-correlation at the corresponding time. These underscore the impact of work schedules and daily routines on traffic dynamics.

As shown in Figure 4, the annual traffic flow trend is heterogeneous across districts, illustrating the economic development and population trends. The flow in D03 acutely reduced in 2006, because PEMS merges some suburban sensors into it. Overall, the trend is flat, except in 2020, when the COVID-19 pandemic caused a drop in traffic flow, which mirrors the major public event's influence on transportation.

**Spatial Dynamic Characteristics.** A key characteristic of DynST is its time-varying road network topology. Figure 5 showcases the increase in node density and new road construction between 2001 and 2016. Moreover, we present the spatial dynamics of DynST through three statistical indicators in Table 2, Figure 6 and Figure 7:

- Node dynamics: The average number of nodes added/removed per day.

- Edge dynamics: The average number of edges added/removed per day.

- Graph lifetime: The average number of days that the graphs maintain the same number of nodes, edges, and connections.

This spatial dynamic characteristic is crucial for training models that can learn generalizable, graph-agnostic patterns, ultimately enhancing their prediction accuracy in the target region.

Table 2: Spatial dynamics.

| Region | Node Dynamics | Edge Dynamics | Graph Lifetime |
|--------|---------------|---------------|----------------|
| D03 | 0.1236 | 0.1251 | 6.9412 |
| D04 | 0.2552 | 0.2577 | 2.2435 |
| D05 | 0.0611 | 0.0611 | 17.9610 |
| D06 | 0.0896 | 0.0898 | 15.0357 |
| D07 | 0.1826 | 0.1840 | 3.3582 |
| D08 | 0.1733 | 0.1753 | 7.4002 |
| D10 | 0.1095 | 0.1092 | 2.0037 |
| D11 | 0.0719 | 0.0730 | 7.7283 |
| D12 | 0.0778 | 0.0789 | 7.8754 |

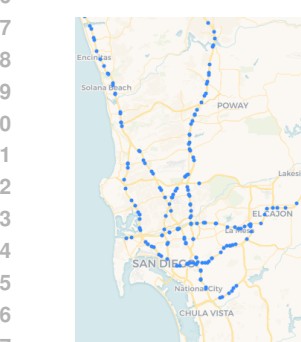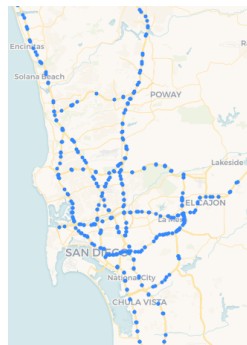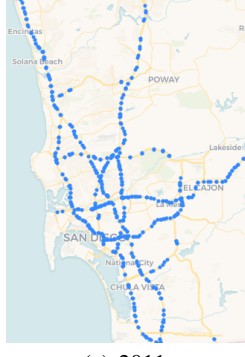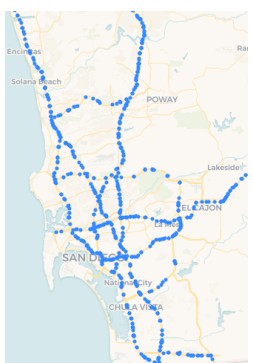

(a) 2001.  (b) 2006.  (c) 2011.  (d) 2016.

Figure 5: Illustration of sensor positions over time. Subfigure (a)-(d), four road networks in the years of 2001-2016, reflect the spatial dynamism of the real-world traffic.

## 5 EXPERIMENT

To verify the necessity and effectiveness of DynST for transfer learning models, we design experiments following mainstream setting (Wang et al., 2021; Yin et al., 2022; Jin et al., 2022; Yao et al., 2019; Jin et al., 2022). A non-transfer learning experiment is supposed as a control group under data-scarce scenarios, but this is challenging in DynST because non-transfer learning models disregard changes in graph structures, while DynST directly uses dynamic topology to characterize regional road networks. Fortunately, over 20 years, there are still certain periods in DynST where the road network remains structurally consistent, meeting the requirements of non-transfer learning models. We refer to this period as the *fixed road network version* of DynST, with specific details provided in the appendix B. Then we conduct transfer learning experiments, and, specifically, according to the source dataset version, categorize them into experiments conducted on the fixed road network version and the dynamic road network version.

Since the main variation across these three experimental groups is the dynamism of spatial topology, we need to align the time spans to ensure the rigor of our experiments. However, the time span of the first experimental group's data is affected by unexpected real-world events, resulting in a monthly duration. To better validate the effectiveness of the dataset, we also include experiments using the native Transformer model over a longer time span for transfer learning.

### 5.1 BASIC SETUP

Following the switching setting of source and target region, we randomly select two districts with medium-sized nodes: D06 and D11. For each district, we used the most recent consecutive 50-day subsets (denoted as D06′ and D11′) from the fixed road network version. The validation and test sets are the same across the 4 experiments, where validation is from the 31st to the 40th day, and the test set is from the 41st to the 50th day. The former 30 days are used as the training set in the target region in Exp.2 – 4 (See more in Appendix F).

Table 3: Performance comparison of Exp.1 that Non-transfer learning with fixed road network.

| Dataset | Method | 1-day | | | 3-day | | | 10-day | | | 30-day | | |
|---------|--------|-------|------|------|-------|------|------|--------|------|------|--------|------|------|
| | | MAE | RMSE | MAPE | MAE | RMSE | MAPE | MAE | RMSE | MAPE | MAE | RMSE | MAPE |
| D06′ | LSTM | 18.03 | 30.64 | 23.68 | 15.78 | 25.94 | 19.22 | 13.35 | 21.54 | 16.71 | **12.64** | **20.12** | **15.65** |
| | STGCN | 21.19 | 34.47 | 25.33 | 15.92 | 26.40 | 19.04 | 12.28 | 19.73 | 14.50 | **11.73** | **18.71** | **14.26** |
| | GWNET | 17.06 | 27.13 | 24.20 | 16.32 | 26.09 | 20.45 | 12.09 | 19.27 | 14.89 | **11.08** | **17.60** | **13.70** |
| | ASTGCN | 17.99 | 28.31 | 24.73 | 15.91 | 26.03 | 18.83 | 13.79 | 21.82 | 17.34 | **12.52** | **19.81** | **15.99** |
| D11′ | LSTM | 29.45 | 45.38 | 21.65 | 23.42 | 36.72 | 20.57 | 19.69 | 32.02 | 13.64 | **18.68** | **30.84** | **13.00** |
| | STGCN | 33.67 | 51.44 | 29.84 | 24.55 | 37.83 | 21.98 | 19.05 | 30.40 | 14.63 | **18.12** | **29.78** | **13.53** |
| | GWNET | 26.99 | 41.67 | 25.33 | 22.89 | 35.73 | 17.64 | 17.99 | 29.11 | 14.07 | **16.93** | **27.73** | **12.73** |
| | ASTGCN | 28.70 | 42.74 | 32.14 | 23.82 | 37.44 | 20.69 | 20.86 | 32.60 | 17.10 | **19.44** | **30.93** | **14.40** |

Table 4: Performance comparison between Exp.2 that Transfer learning with fixed road network and Exp.1. Best: best results of Exp.1.

| Dataset | Setting | Method | 1-day | | | 3-day | | |
|---------|---------|--------|-------|------|------|-------|------|------|
| | | | MAE | RMSE | MAPE | MAE | RMSE | MAPE |
| D06′ | Exp.1 | Best | 17.06 | 27.13 | 23.68 | **15.78** | 25.94 | 18.83 |
| | Exp.2 | ST-GFSL | 17.73 | 30.19 | 23.37 | 17.29 | 27.18 | 21.90 |
| | | DASTNet | **16.78** | **26.01** | **20.44** | 15.9 | **24.7** | **18.82** |
| D11′ | Exp.1 | Best | 26.99 | 41.67 | 21.65 | 22.89 | 35.73 | 17.64 |
| | Exp.2 | ST-GFSL | 25.56 | 40.52 | 19.29 | 24.35 | 38.33 | 18.85 |
| | | DASTNet | **24.11** | **36.72** | **18.2** | **22.62** | **34.39** | **16.82** |

Table 5: Performance comparison between Exp.3 that Transfer learning with dynamic road network and Exp.2.

| Dataset | Setting | 1-day | | | 3-day | | |
|---------|---------|-------|------|------|-------|------|------|
| | | MAE | RMSE | MAPE | MAE | RMSE | MAPE |
| D06′ | Exp.2 | 17.73 | 30.19 | 23.37 | 17.29 | 27.18 | 21.90 |
| | Exp.3 | **17.58** | **30.09** | **21.4** | **16.7** | **26.78** | **20.85** |
| D11′ | Exp.2 | 25.56 | 40.52 | 19.29 | 24.35 | 38.33 | 18.85 |
| | Exp.3 | **25.42** | **40.15** | 19.94 | 24.69 | 38.78 | **18.07** |

We take the mainstream settings of the historical period $T = 12$ and the future period $T' = 12$, with a granularity of 288 steps per day, i.e., using the previous hour to predict the next hour. The samples are preprocessed by Z-score normalization.

We employed three well-established metrics to evaluate model performance: mean absolute error (MAE), root mean squared error (RMSE), and mean absolute percentage error (MAPE). Our experiments were conducted on the NVIDIA GeForce RTX 3090, and the baseline models were benchmarked according to their publicly available code. The hyperparameters are consistent with those in the original paper except for those specifically stated for alignment purposes.

## 5.2 BASELINES

We choose representative traffic forecasting baselines. (1) **LSTM** Hochreiter & Schmidhuber (1997): long-short-term memory, the classical sequential model. (2) **STGCN** Yu et al. (2018): Spatial-Temporal Graph Convolutional Network, which adopts GCN to capture the spatial information. (3) **GWNet** Wu et al. (2019): Graph WaveNet, utilizing the dilated casual convolutions and graph convolutions to capture correlation. (4) **ASTGCN** Guo et al. (2019): Attention Based Spatial-Temporal Graph Convolutional Network, introducing attention mechanism for traffic forecasting. (5) **ST-GFSL** Lu et al. (2022): Spatial-Temporal Graph Few-Shot Learning, proposing a model-agnostic learning framework via node-level meta-learning and parameter matching technique. (6) **DASTNet** Tang et al. (2022): Domain Adversarial Spatial-Temporal Network, employing graph representation learning and adversarial adaptation for multi-city. Notably, DASTNet and ST-GFSL focus on transfer learning.

Table 6: Performance comparison between Exp.4 that long-range transfer learning experiment with dynamic road network and Exp.1 best results.

| | Zero-shot | | | 1-day | | | 3-day | | |
| | MAE | RMSE | MAPE | MAE | RMSE | MAPE | MAE | RMSE | MAPE |
|---|---|---|---|---|---|---|---|---|---|
| Exp.1 Best | – | – | – | 26.99 | 41.67 | 21.65 | 22.89 | 35.73 | 17.64 |
| w/o pretrained | – | – | – | 28.66 | 44.82 | 23.33 | 24.2 | 38.16 | 19.74 |
| 1-year | 26.34 | 42.70 | 21.79 | 23.97 | 39.36 | 19.39 | 21.90 | 35.64 | 17.11 |
| 5-year | 26.74 | 44.14 | 21.76 | 23.62 | 38.99 | 17.78 | 20.74 | 33.70 | 15.21 |
| 20-year | **26.31** | **42.62** | **18.93** | **23.57** | **38.58** | **17.07** | **20.24** | **32.98** | **14.92** |

## 5.3 Effectiveness of DynST

**Exp.1: Non-transfer learning with fixed road network.** As a control group, the models in this experiment are trained and tested in the same region of the fixed road network version. This setting simulates a data-scarcity scenario where the available data grows over time, and the training size increases from a 1-day to a 30-day time range. Table 3 presents the experimental results. As observed, the performance of all models consistently improves with a gradual increase in the training set size.

**Exp.2: Transfer learning with a fixed road network.** This experiment aligns with the traditional setting of transfer learning tasks, where the model is trained on the source region with a fixed road network, and then the model is transferred to the target region for prediction. Following previous studies (Wang et al., 2021; Tang et al., 2022; Lu et al., 2022), we randomly choose one out of the two districts as the source region, while the remaining district serves as the target region. To simulate real-world data scarcity scenarios, we employ a limited training set size for the target region, 1-day and 3-day. The source region leverages an extensive training set size of 30 days. The baseline models chosen for this evaluation are DASTNet (Tang et al., 2022) and ST-GFSL (Lu et al., 2022). To provide a comprehensive comparison, we show the best-performing results excerpted from Exp.1 in Table 3.

The results are shown in Table 4. DASTNet generally outperforms the best results of Exp.1, which verifies the effectiveness of the transfer learning technique. While ST-GFSL demonstrates improvements in part of evaluation metrics, its performance might be limited by the expressive power of the backbone model GRU (Chung et al., 2014), which struggles with capturing complex dependencies within traffic data sequences.

**Exp.3: Transfer learning with a dynamic road network.** This experiment is a brand new transfer learning setting for dynamic road network topology. The training set of the source region is the dynamic version of DynST. As far as we know, ST-GFSL (Lu et al., 2022) is the only model that can, in principle, handle the infinite number of evolving road networks without modification to the model architecture. We follow the default hyperparameters reported in the original paper (Lu et al., 2022) and turn the size of the dynamic road network sampling pool to align with the time span of Exp.1 – 2.

The results are shown in Table 5. ST-GFSL boosts the performance compared to one of Exp.2 settings in most scenarios, which inspires us to develop an elaborate design to better capture rich information of the spatial dynamism in the source region.

**Exp.4: Long-range transfer learning experiment with dynamic road network.** Although Exp.1–3 have demonstrated the effectiveness of spatial dynamics provided by the dynamic road network in DynST, the data points' usage is limited to a monthly duration due to the alignment with temporal dynamics. Since the dynamic characteristics of the road network need to be reflected over time, we ease the time length restriction in Exp.4 to showcase the effectiveness of the 20-year data volume.

We conducted experiments with longer time spans with dynamic road networks, as shown in Table 6. An interesting episode is that the ST-GFSL exhibited loss spikes, ultimately leading to model collapse. We use Transformer (Vaswani et al., 2017) as the baseline without any transfer technique. We pre-train it on full-volume D06, and test performance on D11'. The best results from the non-transfer learning methods in Exp.1 are taken for comparison. We gradually increase the amount of pre-training data from no pre-training to 1 year, 5 years, and then 20 years, and compare the results of zero-shot and those using 1-day and 3-day fine-tuning data volume. The zero-shot results pre-trained on 20 years of data volume even outperform the best results under 1-day volume of Exp.1 in both MAE and MAPE. This demonstrates that, with the support of large-scale dynamic road network data in DynST, the model has already shown a certain degree of generalization ability in the target region, even without

Table 7: The comparison between the tree-based (ours) and distance-based adjacency matrix methods, which are denoted as method-t and method-d, respectively.

| Method | 1-day | | | 3-day | | | 10-day | | | 30-day | | |
|---|---|---|---|---|---|---|---|---|---|---|---|---|
| | MAE | RMSE | MAPE | MAE | RMSE | MAPE | MAE | RMSE | MAPE | MAE | RMSE | MAPE |
| STGCN-d | 21.24 | 34.58 | 25.65 | 15.87 | 26.21 | 18.26 | 12.51 | 20.26 | 14.88 | 11.74 | 18.62 | 14.35 |
| STGCN-t | **21.19** | **34.47** | **25.33** | 15.92 | 26.40 | 19.04 | **12.28** | 19.73 | **14.50** | **11.73** | 18.71 | **14.26** |
| GWNET-d | 17.12 | 27.20 | 25.93 | 16.44 | 26.28 | 21.50 | 12.04 | 18.99 | 16.54 | 11.27 | 17.84 | 14.30 |
| GWNET-t | **17.06** | **27.13** | **24.2** | **16.32** | **26.09** | **20.45** | 12.09 | 19.27 | **14.89** | **11.08** | **17.60** | **13.7** |
| ASTGCN-d | 18.41 | 28.84 | 25.32 | 18.84 | 25.62 | 18.80 | 13.23 | 21.30 | 16.24 | 12.57 | 20.06 | 15.24 |
| ASTGCN-t | **17.99** | **28.31** | **24.73** | **15.91** | 26.03 | 18.83 | 13.79 | 21.82 | 17.34 | **12.52** | **19.81** | 15.99 |
| STGCN-d | 33.29 | 50.79 | 30.57 | 24.18 | 37.18 | 22.11 | 19.10 | 30.59 | 14.33 | 18.05 | 29.63 | 13.37 |
| STGCN-t | 33.67 | 51.44 | **29.84** | 24.55 | 37.83 | **21.98** | 19.05 | **30.4** | 14.63 | 18.12 | 29.78 | 13.53 |
| GWNET-d | 27.00 | 41.59 | 25.68 | 23.81 | 35.71 | 25.36 | 17.94 | 28.98 | 13.38 | 16.61 | 27.38 | 13.43 |
| GWNET-t | **17.06** | **27.13** | **24.2** | **16.32** | **26.09** | **20.45** | **12.09** | **19.27** | 14.89 | **11.08** | **17.60** | 13.7 |
| ASTGCN-d | 29.51 | 43.87 | 32.63 | 23.26 | 36.60 | 20.13 | 22.03 | 46.72 | 19.76 | 21.44 | 38.82 | 19.07 |
| ASTGCN-t | **17.99** | **28.31** | **24.73** | **15.91** | **26.03** | **18.83** | **13.79** | **21.82** | **17.34** | **12.52** | **19.81** | **15.99** |

(Row groups labeled D06′ for the first six rows and D11′ for the last six rows.)

the need for specialized transfer learning techniques. Although we have not yet seen capabilities comparable to those of foundational models in NLP and CV, it does exhibit promising potential in the field of traffic forecasting.

## 5.4 Effectiveness of Adjacency Matrix Generation Method

The guiding principle of the tree-based adjacency matrix generation algorithm we propose is to closely resemble the real road network situation. However, it is worth noting that a well-structured road network should assist baselines in capturing the complex dependencies within the network. We constructed a new adjacency matrix based on the distance-based method proposed by largeST (Liu et al., 2023b) and re-conducted Exp.1. The results are shown in table 7. It can be observed that the tree-based adjacency matrix generally led to better results for the models. Notably, in the D11′, GWNet and ASTGCN achieved significant improvements using the tree-based adjacency matrix. These results demonstrate the effectiveness of our tree-based adjacency matrix generation method in helping the model learn spatial dependencies.

## 6 Conclusion

We present a large-scale dataset, DynST, designed for transfer learning tasks in traffic forecasting. The key distinction of DynST from existing datasets lies in its adoption of an evolving dynamic road network topology, which better captures the construction evolution of regional road networks and the impact of unexpected events. DynST encompasses 20.35 billion data points collected over 20 years across 9 regions in California. In addition to providing traffic signal data and metadata, we propose a tree-based adjacency matrix generation algorithm that better aligns with the upstream and downstream connection relationships of real road networks. We conducted extensive statistical analyses to aid in understanding the temporal and spatial characteristics of DynST. Experiments demonstrate that the dynamic spatial topology provided in DynST facilitates better transfer and generalization of models to the target region. Compared to the distance-based adjacency matrix generation algorithm, our proposed tree-based method shows greater benefits for model prediction. The pretraining experiment conducted over the full time span reveals that the baseline model exhibits a certain degree of zero-shot capability. We hope that DynST will provide insights to the community and promote the emergence of foundational models in the spatial-temporal prediction field.

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

## A  DETAILS OF DATA CLEANING

For data cleaning, there are many practical considerations. For dynamic missing values in raw data, there are generally two handling strategies. The first is to discard the missing values. While some sensors provide high-quality data and can operate continuously, many other sensors often experience significant data loss over certain days. If we only retain high-quality points, it will drastically reduce the size of the road networks and the time span. The second strategy is interpolation, which inevitably disrupts data dependencies. Some studies (Guo et al., 2023; Chen et al., 2001) have pointed out that traffic data distribution exhibits specific abrupt changes, and the data produced by interpolation lacks such variation, which can severely impact model learning.

We adopt a daily processing granularity, ensuring that the road network topology remains the same within a day. We discard points with missing values that occur on that day. This strategy ensures both data volume and data reliability.

## B  FIXED AND DYNAMIC ROAD NETWORK VERSION

DynST uses a dynamic road network topology to describe the real road network, which means that the road network structure in the dataset changes over time. Non-transfer learning models cannot handle this kind of change, but we still need Exp.1 for comparison. Fortunately, over 20 years, there have been instances of the road network remaining unchanged in DynST. We compared the road networks for two consecutive days, and when they were completely identical, we merged them, ultimately resulting in this fixed road network version of DynST.

Thereby, we provide two versions of DynST, and the fixed one can be gained by running the merging script to process the dynamic version.

## C  STORAGE

We offer two versions of DynST. The dynamic version is available on OneDrive, totaling 78.6 GB, and compressed in the `npz` format of NumPy [3]. This version includes two file packages: one containing the feature data stored in `np.float16` format, and the other containing the Sensor IDs stored as `np.int32`.

For the fixed version, users can generate the dataset by using the provided `merge.py` script. This script merges data points across the temporal dimension when the road network graphs in a continuous timeline are identical.

## D  METADATA DETAIL

We provide metadata grouped by district, with detailed descriptions available in Table 8. We believe that offering this metadata will facilitate a research fine-grained in the future.

Similar to how our node and edge construction processes operate on a daily basis, metadata is also processed daily. The metadata reflects on the infrastructure development and neglects the sudden events, which are reflected by the nodes and edges filtered by rules. To efficiently manage storage and prevent redundant data, we employ a mechanism to store only metadata that differs from the previous day. This includes changes in sensor count, deployment locations, or other critical attributes of the road network. So in the data repository, metadata updates irregularly.

---

[3] https://numpy.org/

Table 8: The description of metadata attribution.

| Metadata | Description |
|---|---|
| ID | An integer value that uniquely identifies the Sensor. |
| Freeway | Freeway Number |
| Direction | A string representing the freeway direction. |
| County | The unique number that identifies the county. |
| City | The city that the freeway belongs to. |
| State Postmile | State postmile. |
| Absolute Postmile | Absolute postmile. |
| Latitude | Latitude. |
| Logitude | Longitude. |
| Length | Length. |
| Type | The type of the sensor. |
| Lanes | Total number of lanes. |

# E DETAILS OF ADJACENCY MATRIX GENERATION ALGORITHM

To construct the adjacency matrix, we utilize the shortest path algorithm and the minimum spanning tree algorithm. Here, we present the complete pseudo-code for the shortest path algorithm using Dijkstra's AlgorithmDijkstra (1959) (Algorithm 2) and the minimum spanning tree algorithm using Kruskal's AlgorithmKruskal (1956) (Algorithm 3).

Regarding our degree of node prior knowledge, we give the following analysis. DynST and the existing PEMS datasets come from the same collection system PEMS. The PEMS system collects traffic signals on expressways and highways. Through visual analysis in Figure 5, it can be observed that there are mostly straight roads (single chain graphs) supplemented by a few crossings. The degree of the nodes in the single chain graph and the degree at the crossings are both close to 1 (with 5 points and 4 edges forming a crossing). Therefore, we believe that when considering larger graphs including remote areas, the overall degree will be approximately 1.0. By conducting a provenance analysis, we found that the existing PEMS dataset's sensors are mainly concentrated in urban areas, where there are more complex situations, resulting in a higher degree (PEMS03 1.5 and PEMS08 1.6). In contrast, our distribution includes many towns and remote areas, where straight roads are more common. Overall, our degree will be lower than that of the existing PEMS dataset, approaching the theoretical value of 1.0.

---

**Algorithm 2:** shortestPathLength

---

**Input** : A weighted graph $\mathcal{G}_d = (\mathcal{V}_d, \mathcal{E}_d)$, source vertex $u$, destination vertex $v$
**Output :** The shortest path length from $u$ to $v$

---

1 create vertex set $\mathcal{Q}$
2 **foreach** vertex $v \in \mathcal{G}_d$ **do**
3     $dist[v] \leftarrow \infty$
4     add $v$ to $\mathcal{Q}$
5 $dist[source] \leftarrow 0$
6 **while** $\mathcal{Q}$ is not empty **do**
7     $u \leftarrow$ vertex in $\mathcal{Q}$ with min $dist[u]$
8     remove $u$ from $\mathcal{Q}$
9     **foreach** neighbor $v$ of $u$ still in $\mathcal{Q}$ **do**
10        $alt \leftarrow dist[u] + \text{length}(u, v)$
11        **if** $alt < dist[v]$ **then**
12           $dist[v] \leftarrow alt$

13 **return** $dist[v]$

---

---

**Algorithm 3:** minimumSpanningTree

---

**Input** : A weighted graph $\mathcal{G}_d = (\mathcal{V}_d, \mathcal{E}_d)$
**Output** : A minimum spanning tree $\mathcal{F}$ of $\mathcal{G}_d$

1   $\mathcal{F} \leftarrow \emptyset$
2   **foreach** vertex $v$ in $\mathcal{V}_d$ **do**
3     MAKE-SET($v$)
4   **foreach** edge $(u, v)$ in $\mathcal{E}_d$ ordered by weight $(u, v)$, increasing **do**
5     **if** FIND-SET($u$) $\neq$ FIND-SET($v$) **then**
6       $\mathcal{F} \leftarrow \mathcal{F} \cup \{(u, v)\}$
7       UNION(FIND-SET($u$), FIND-SET($v$))
8   **return** $\mathcal{F}$

---

## F   Details of Experimental Setup

We conducted effectiveness experiments under 4 different settings. To ensure a fair comparison, we randomly selected two districts, D06 and D11. For each district, we used the most recent consecutive 50-day subsets from the fixed version of DynST, denoted as D06′ and D11′. The preprocessing procedure was consistent across all experiments. We split the data into training, validation, and test sets with a ratio of 6:2:2. Specifically, the validation set consisted of data from the 31st to the 40th day, and the test set included data from the 41st to the 50th day. The first 30 days of data are used as the training set in the source region. Detailed information is presented in Table 9.

Table 9: Detailed information of D06′ and D11′.

| Dataset | Nodes | Time Range | | |
| --- | --- | --- | --- | --- |
| | | Training set | Validation set | Test set |
| D06′ | 614 | 01/20/2024 – 02/18/2024 | 02/19/2024 – 02/28/2024 | 02/29/2024 – 03/09/2024 |
| D11′ | 830 | 01/20/2024 – 02/18/2024 | 02/19/2024 – 02/28/2024 | 02/29/2024 – 03/09/2024 |

## G   Challenge and Future Work

DynST has greater potential for transfer learning, it also demands higher requirements for model design. An interesting observation from our preliminary experiments is that the ST-GFSL exhibited loss spikes on DynST of the same length, ultimately leading to model collapse. This further illustrates the challenges posed by DynST for transfer learning, and subsequent research will need to design new transfer learning approaches tailored for DynST. We believe potential approaches could include:

- More flexible node index encoding: STID (Shao et al., 2022) has demonstrated that distinguishing between node indexes and timestamps is essential for the model. Previously, nodes had fixed indexes in the data matrix, which could be encoded using embedding techniques. However, in DynST, due to the changing node, similar simple encoding methods are not feasible. Furthermore, unseen nodes in downstream tasks will need to be encoded to align with nodes from the pre-training phase for better transferring.

- Robust multi-scale network structures: As shown in the accompanying figure in the PDF materials, the number of nodes and edges varies significantly over the years. The model needs to better handle the dependencies between large and small graphs simultaneously. Another possible scenario is that the model relies on relevant points to make predictions for the target node, but at a certain moment, the data from those dependency points may dynamically be lost. This poses a greater requirement for the model's robustness, as it should not overly depend on specific points.

## H    SOCIETY IMPACT

Our work focuses on developing a dataset and exploring transfer learning techniques. These advancements are purely for traffic forecasting and have no inherent negative societal impacts. The data itself is anonymized and focuses solely on traffic flow patterns, posing no privacy concerns.

## I    LIMITATION

With the dynamic evolution of traffic road networks over time, the patterns from 20 years ago may not directly benefit recent predictions. We deem that the routine setting of forecasting 12 steps based on the previous 12 steps, ensures a strong correlation within the sample pair. While the extensive time coverage enriches the dataset with diverse patterns, issues like pattern conflicts and useful information selection are reserved for future research. Researchers have the flexibility to select specific time ranges according to their particular needs.

Based on the investigation, it is inferred that the existing datasets, apart from LargeST, have their adjacency matrices manually labeled by real maps. This approach is impractical for large datasets like DynST. While tree-based adjacency matrix generation algorithms show significant improvements compared to distance-based methods, there will always be discrepancies in real-world road network conditions. We acknowledge this and welcome the emergence of better automatic labeling algorithms.

PEMS is an ongoing project and our data collection work was completed by April 2024. We are considering implementing an automated data updating mechanism in the future.

## J    EXTRA TABLES AND FIGURES

To help users better understand the dataset's characteristics, we have conducted several statistical analyses and generated corresponding charts. We present all the statistical analysis results here.

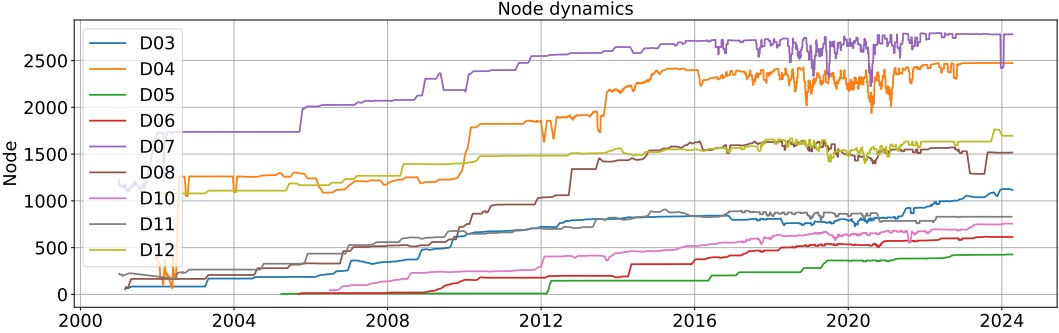

Figure 6: Node Dynamics.

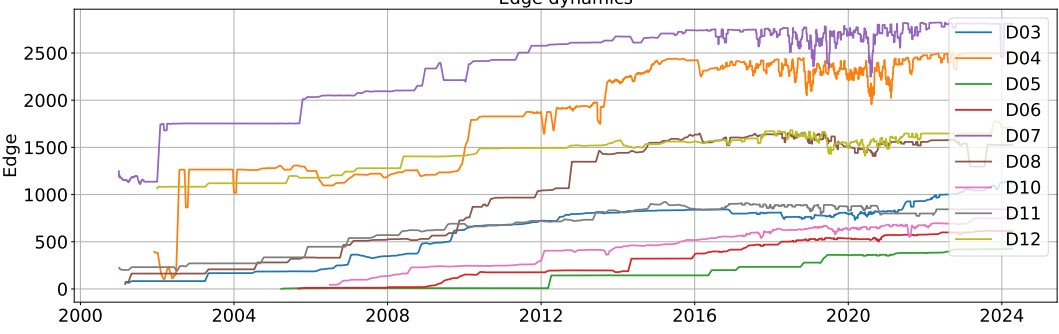

Figure 7: Node Dynamics.

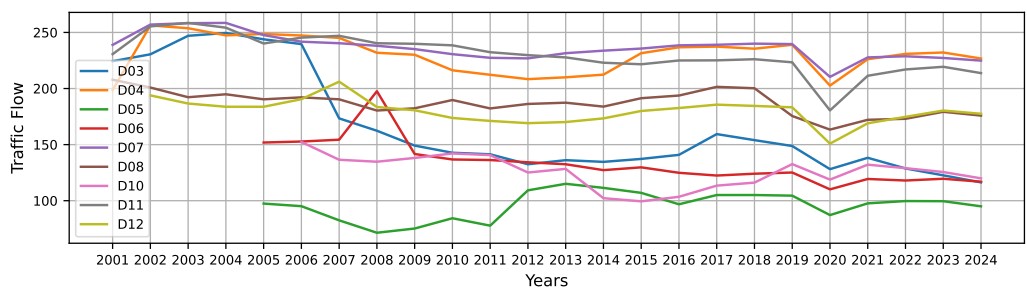

Figure 8: Annual trend of average traffic flow.

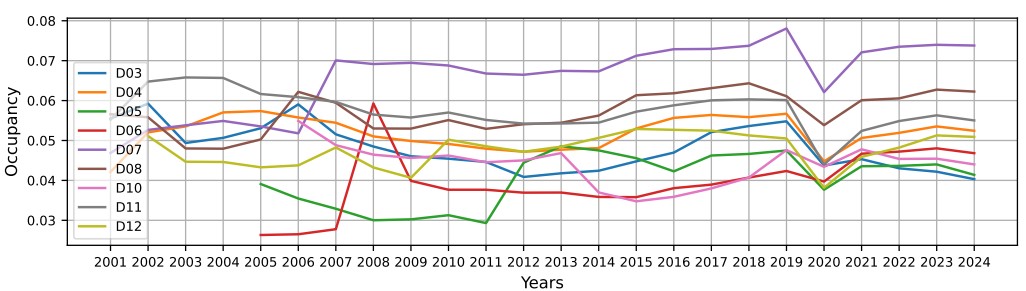

Figure 9: Annual trend of average occupancy.

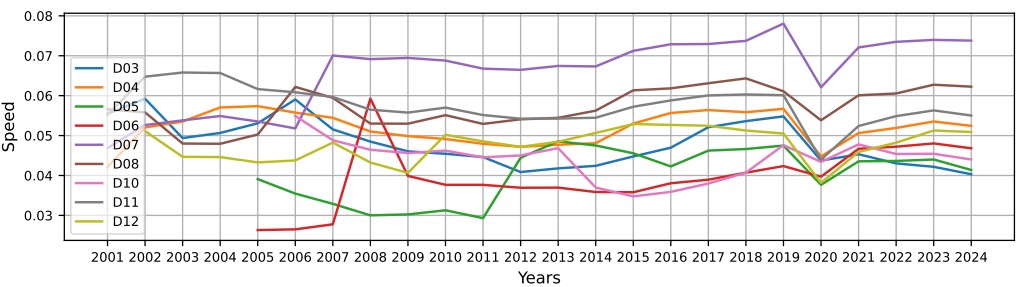

Figure 10: Annual trend of average speed.

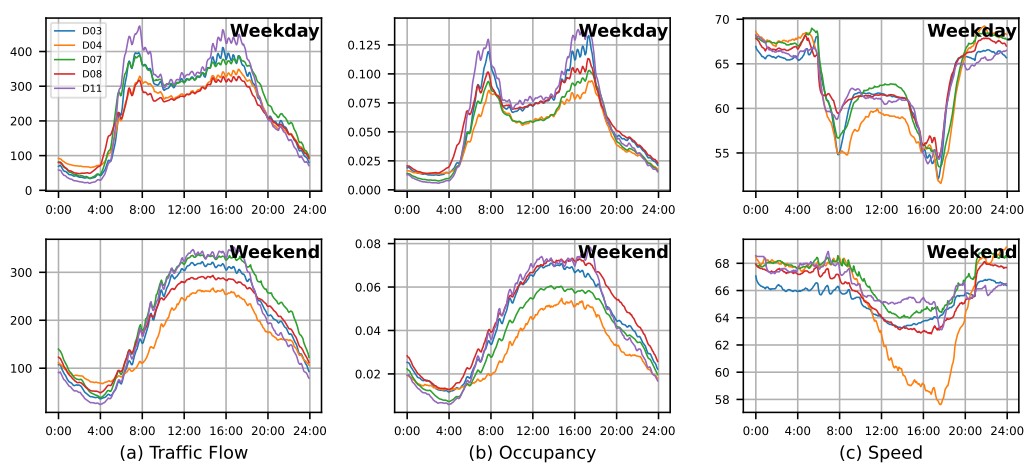

Figure 11: Illustration of weekly patterns of traffic signal distribution in 2001.

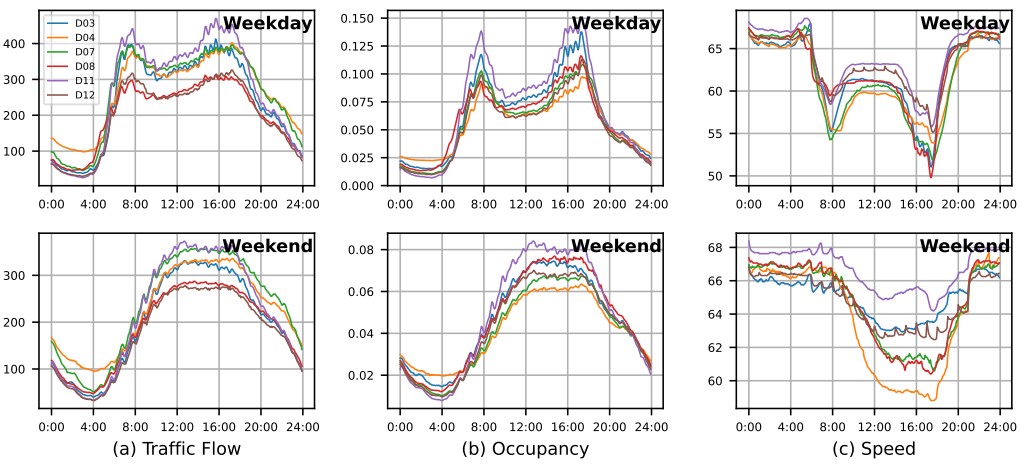

Figure 12: Illustration of weekly patterns of traffic signal distribution in 2002.

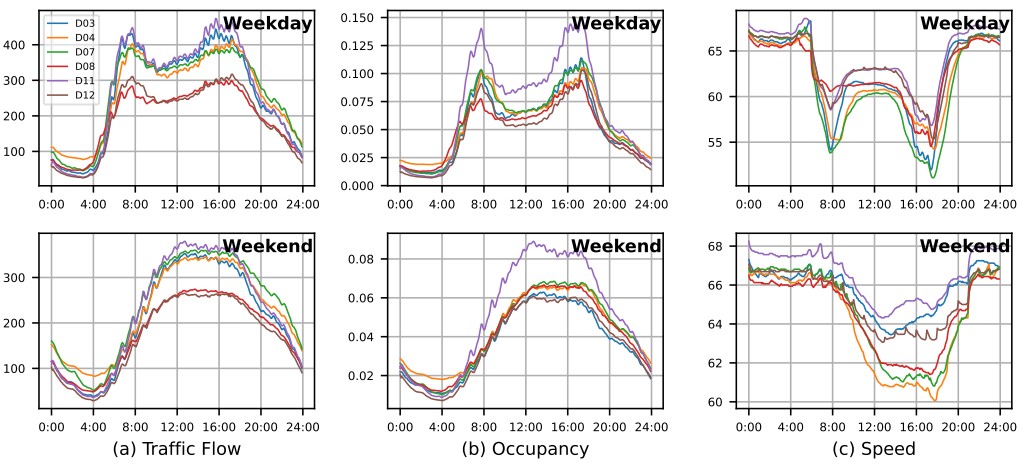

Figure 13: Illustration of weekly patterns of traffic signal distribution in 2003.

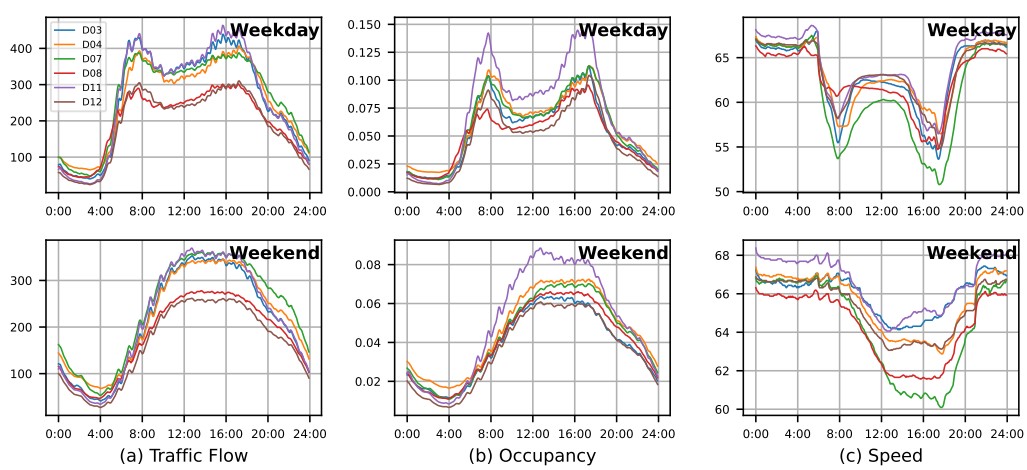

Figure 14: Illustration of weekly patterns of traffic signal distribution in 2004.

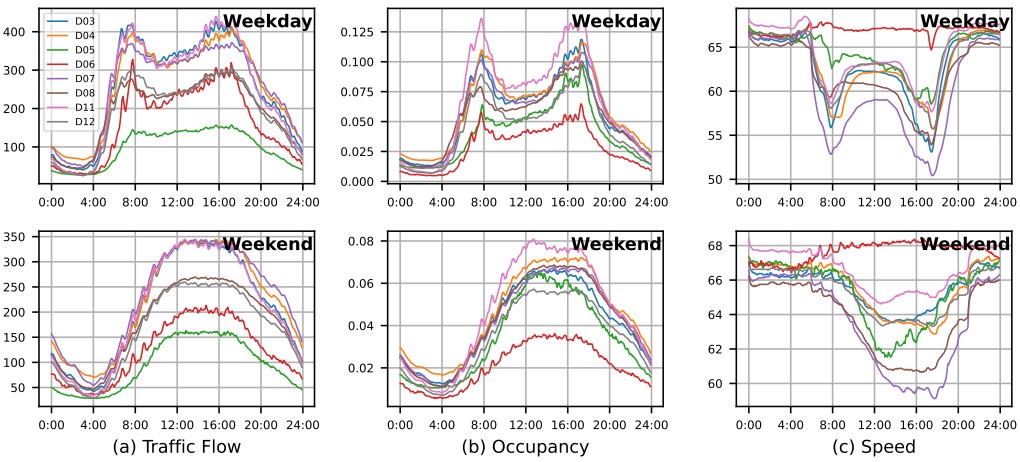

Figure 15: Illustration of weekly patterns of traffic signal distribution in 2005.

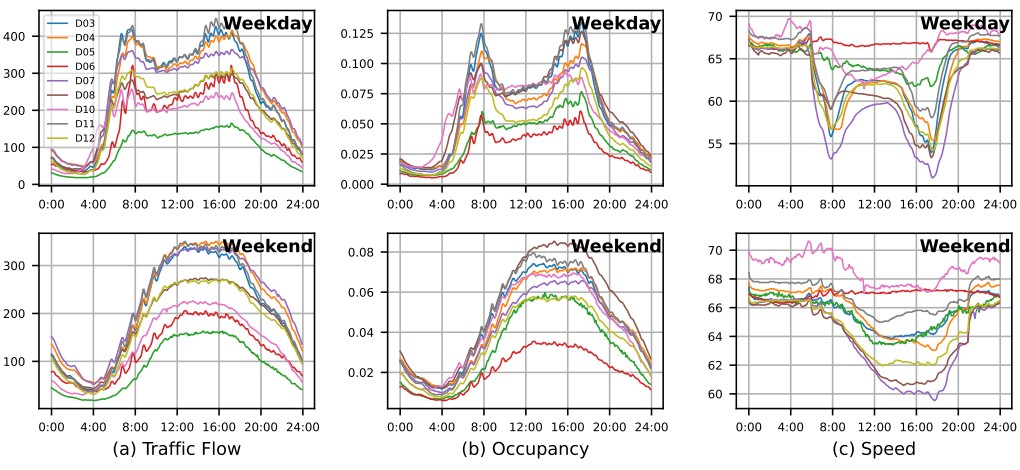

Figure 16: Illustration of weekly patterns of traffic signal distribution in 2006.

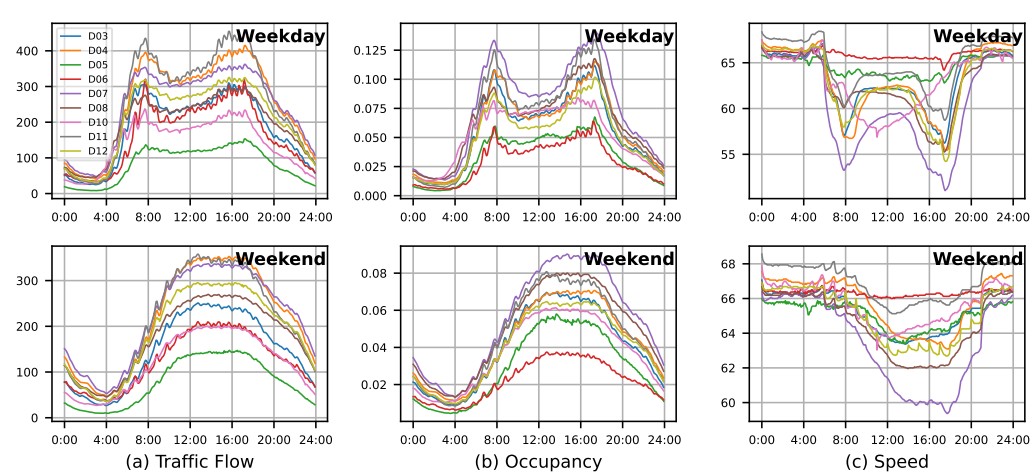

Figure 17: Illustration of weekly patterns of traffic signal distribution in 2007.

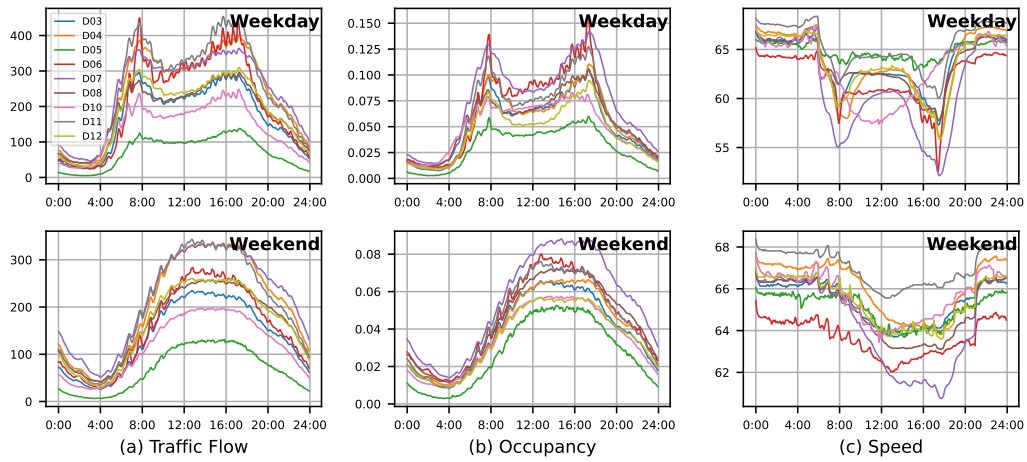

Figure 18: Illustration of weekly patterns of traffic signal distribution in 2008.

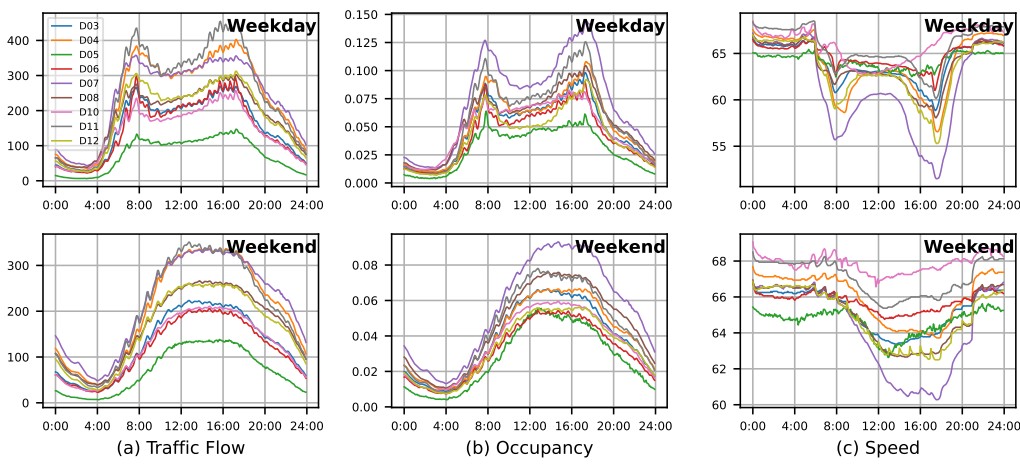

Figure 19: Illustration of weekly patterns of traffic signal distribution in 2009.

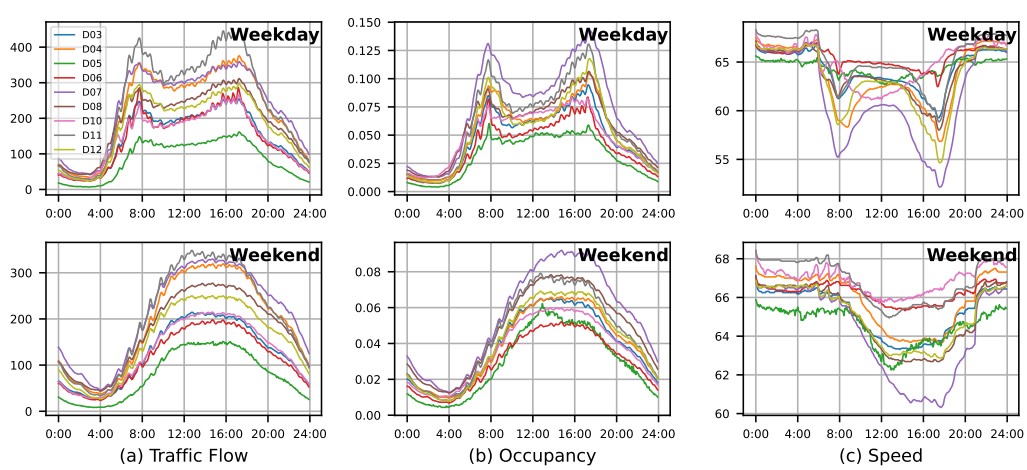

Figure 20: Illustration of weekly patterns of traffic signal distribution in 2010.

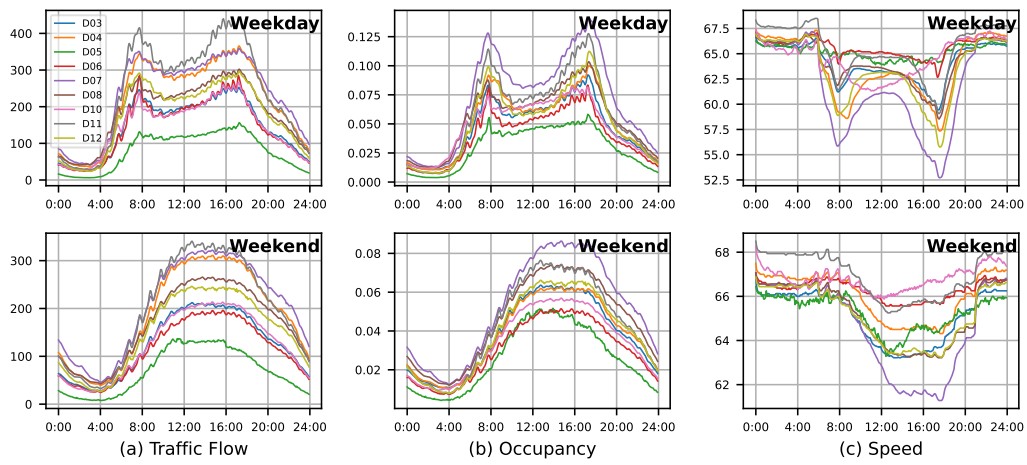

Figure 21: Illustration of weekly patterns of traffic signal distribution in 2011.

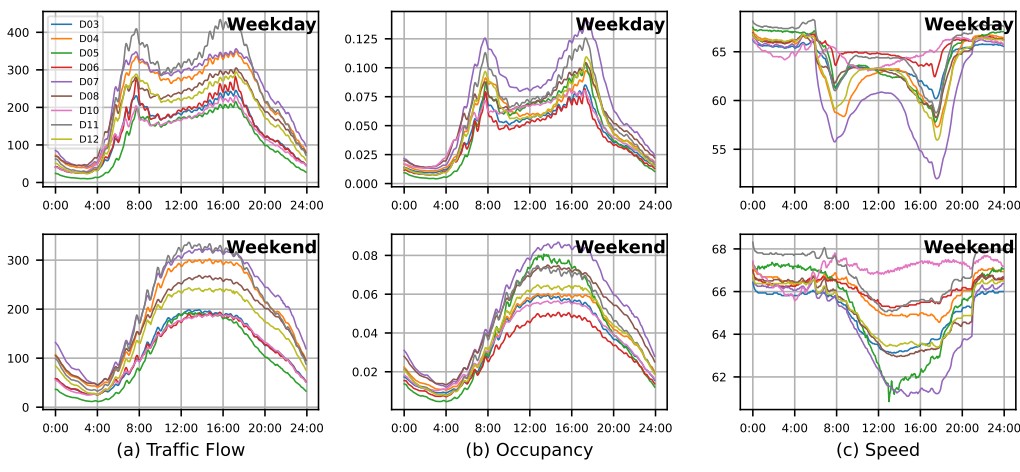

Figure 22: Illustration of weekly patterns of traffic signal distribution in 2012.

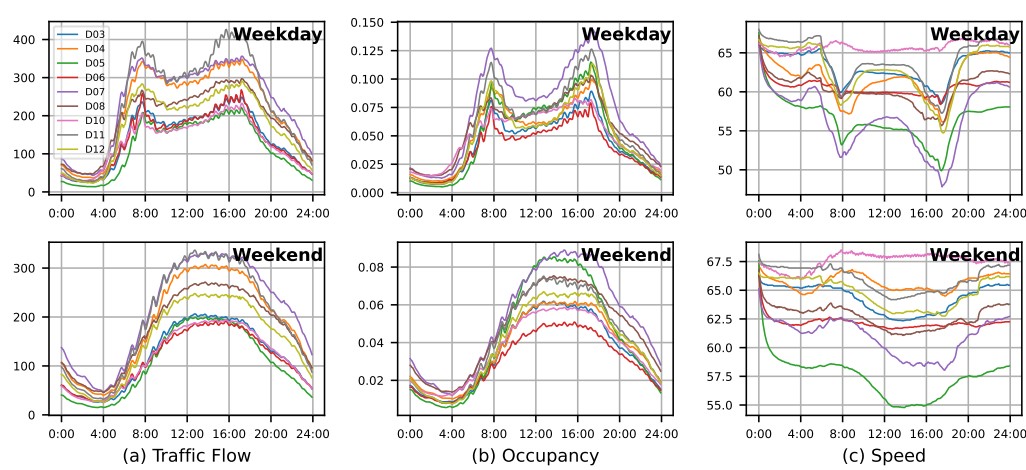

Figure 23: Illustration of weekly patterns of traffic signal distribution in 2013.

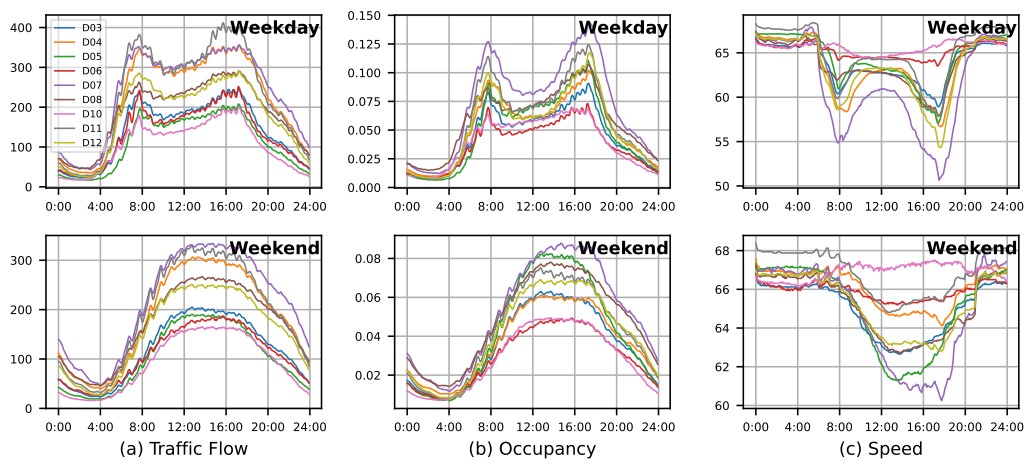

Figure 24: Illustration of weekly patterns of traffic signal distribution in 2014.

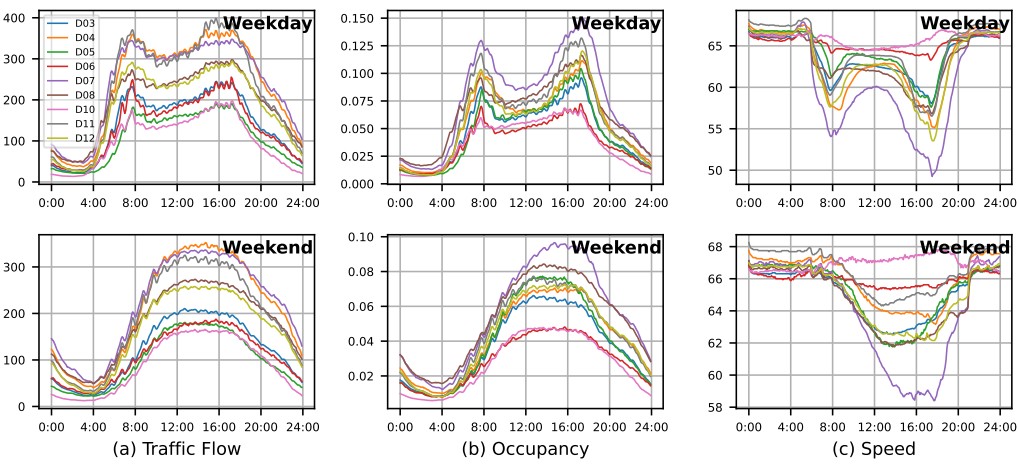

Figure 25: Illustration of weekly patterns of traffic signal distribution in 2015.

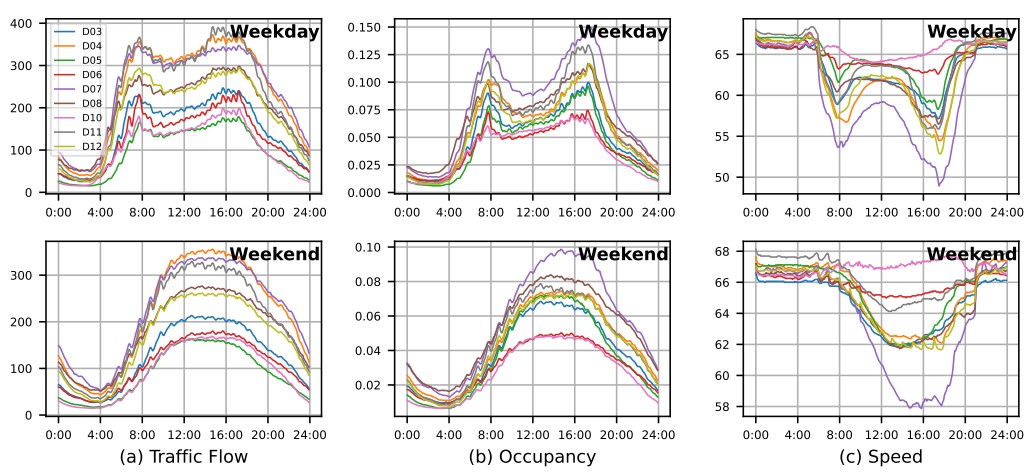

Figure 26: Illustration of weekly patterns of traffic signal distribution in 2016.

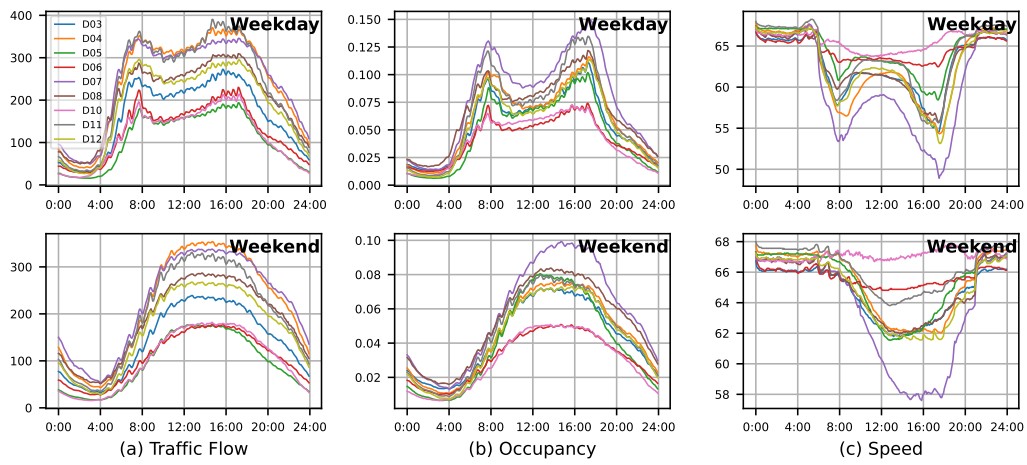

Figure 27: Illustration of weekly patterns of traffic signal distribution in 2017.

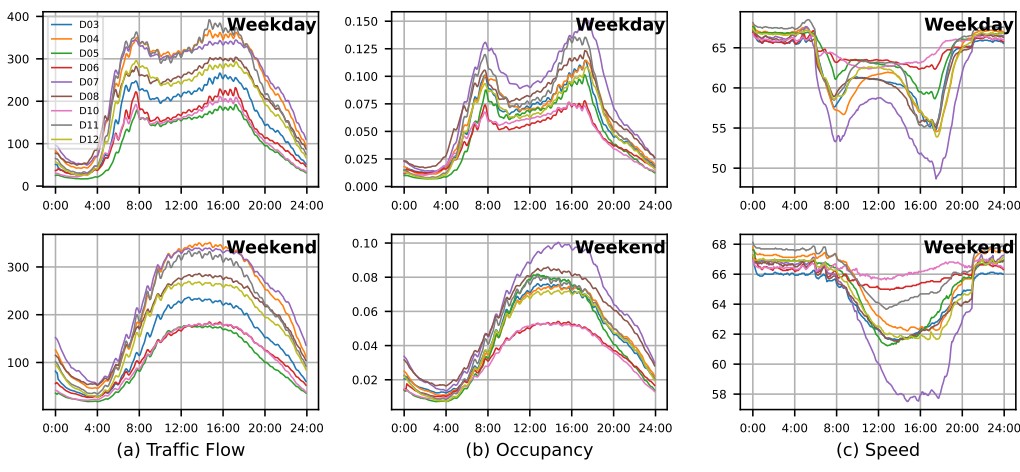

Figure 28: Illustration of weekly patterns of traffic signal distribution in 2018.

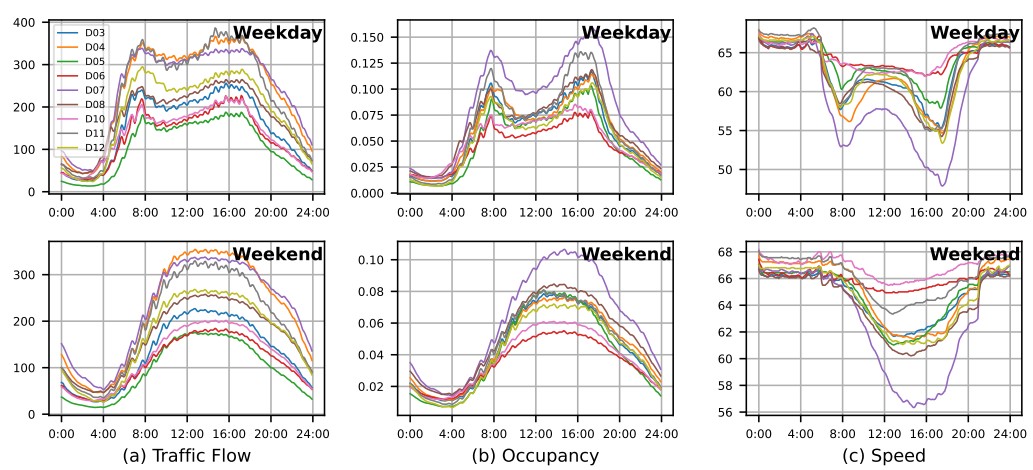

Figure 29: Illustration of weekly patterns of traffic signal distribution in 2019.

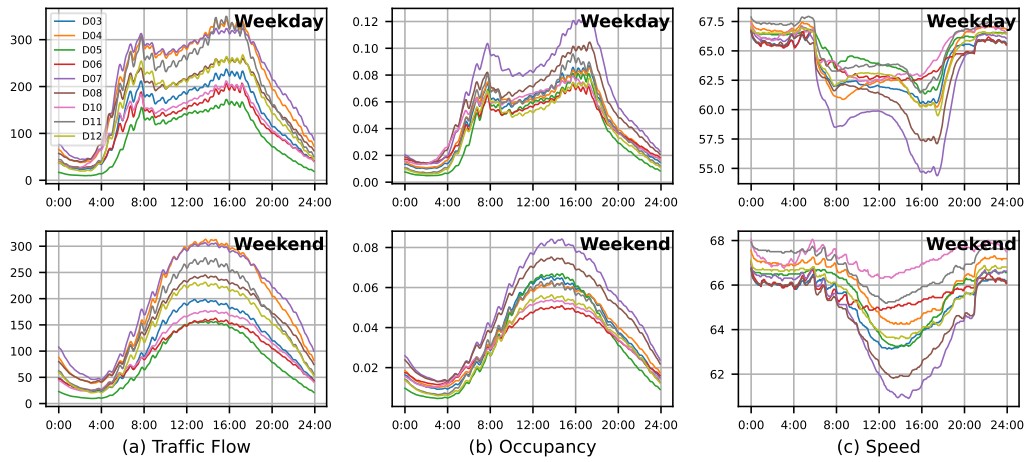

Figure 30: Illustration of weekly patterns of traffic signal distribution in 2020.

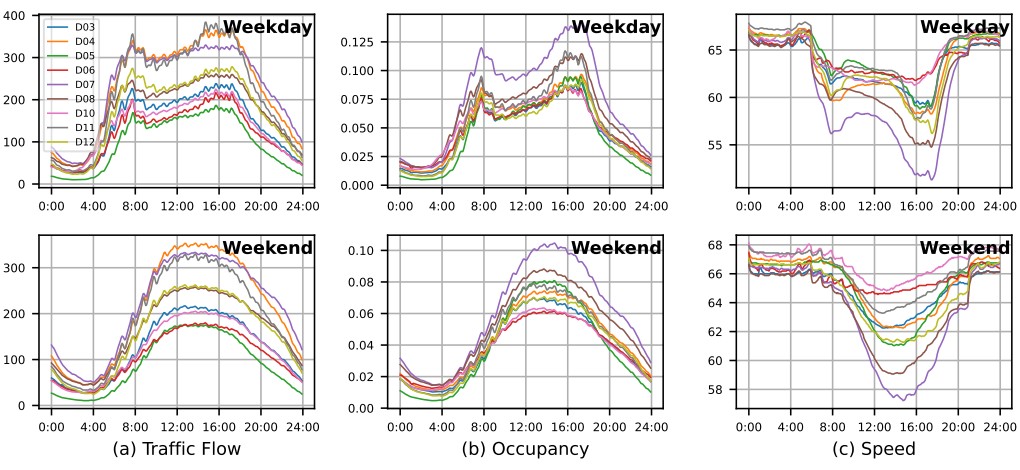

Figure 31: Illustration of weekly patterns of traffic signal distribution in 2021.

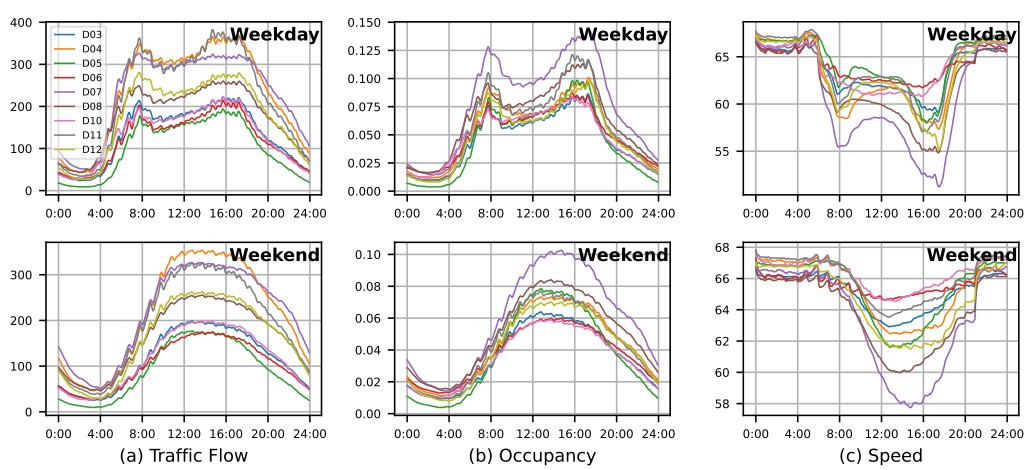

Figure 32: Illustration of weekly patterns of traffic signal distribution in 2022.

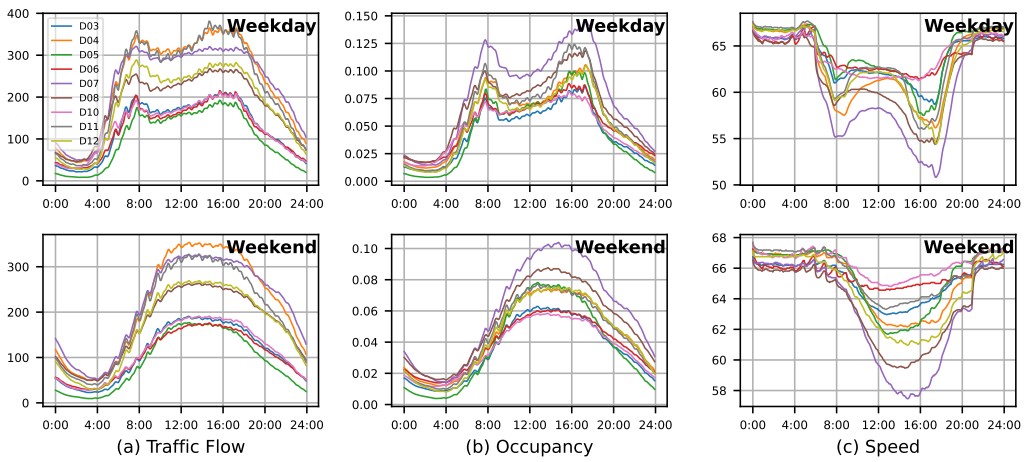

Figure 33: Illustration of weekly patterns of traffic signal distribution in 2023.

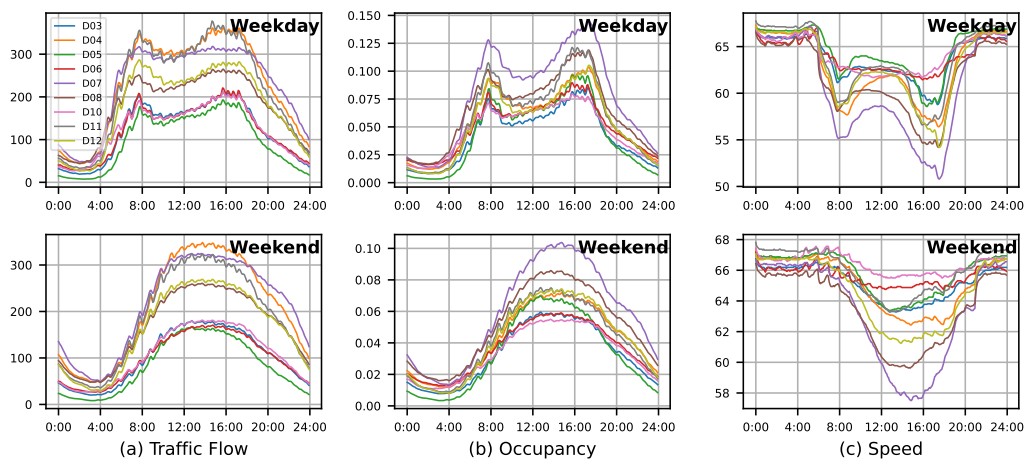

Figure 34: Illustration of weekly patterns of traffic signal distribution in 2024.

