# OpenReview forum: "DynST: Large-Scale Spatial-Temporal Dataset for Transferable Traffic Forecasting with Dynamic Road Networks"
_ICLR.cc/2025/Conference — ICLR 2025 Conference Withdrawn Submission_

### Official Review · Reviewer_Z5Ae · 2024-10-21

**Soundness:** 2
**Presentation:** 2
**Contribution:** 2
**Rating:** 5
**Confidence:** 4

**Summary:**

This paper introduces DynST, a dataset specifically designed for transferrable traffic forecasting. DynST features an evolving road network topology over a 20-year period, covering nine regions and providing over 20 billion data points. To address the overconnectivity and disconnection issues in existing distance-based adjacency generation methods, the paper proposes a tree-based algorithm to construct graph topology that more accurately reflects real-world road connections. Experimental results demonstrate the effectiveness of DynST dataset in cross-region traffic forecasting.

**Strengths:**

1. DynST addresses the gap in generalizable traffic forecasting by providing a tailored dataset, offering potential value to the community.
2. The proposed tree-based adjacency matrix generation algorithm effectively resolves the overconnectivity and disconnection issues that arise in existing methods.
3. The experiments conducted in the paper offer important insights into existing traffic forecasting research, especially in data-scarce scenarios.

**Weaknesses:**

1. The technical contribution of this paper is weak. The proposed road network topology generation algorithm is an extension of minimum spanning tree. The primary contribution is the DynST dataset, which may not well align with the expectations for technical novelty at ICLR.
2. The claim made in this paper that "transfer learning tasks in traffic prediction currently lack dedicated datasets and instead rely on datasets designed for non-transfer prediction tasks" seems inaccurate. In fact, there are several multi-city traffic datasets that are widely used for cross-city transfer learning research. This statement overlooks existing efforts in the community that already address this issue.
3. The motivation behind building such a vast traffic dataset spanning 20 years for transfer learning is not clearly justified. For example, the paper does not adequately explain the benefits or rationale for employing such extensive data for transfer learning.
4. From Table 6, it can be seen that expanding the training data from 1 year to 20 years only provides marginal improvements in the model performance in cross-region prediction scenarios. For instance, the MAE (Mean Absolute Error) improves from 26.34 to 26.31 in the zero-shot setting, and from 21.90 to 20.24 in the 3-day setting, which are relatively small gains considering the significant increase in data volume. While the authors emphasize the scale of DynST as a major contribution, the necessity of utilizing such an enormous dataset is questionable.

**Questions:**

Please see in weaknesses.

---

### Official Review · Reviewer_YkKm · 2024-10-22

**Soundness:** 3
**Presentation:** 3
**Contribution:** 3
**Rating:** 5
**Confidence:** 4

**Summary:**

The paper introduces a dataset named DynST, specifically designed for transfer learning tasks in traffic prediction. DynST comprises an extensive data volume of 20.35 billion entries, spanning 20 years across 9 regions. The evolving dynamic road network topology in DynST reflects the actual development of road networks. To overcome the limitations of  the conventional distance-based adjacency generation algorithm, the paper presents  a novel tree-based algorithm. Extensive experiments indicate that the adoption of DynST as the source dataset can significantly improves the performance of the target region.

**Strengths:**

- The paper introduces a dataset named DynST, which consists of an extensive data volume of 20.35 billion records, spanning 20 years and covering 9 regions. The breadth and depth of this dataset are highly beneficial for training transfer learning models.
- The dataset DynST incorporates the dynamic road network structure, which is designed for transfer learning models, and can reflect the generalization ability of the models.
- The proposed tree-based adjacency matrix generation algorithm can generate topologies that more accurately reflect real-world road networks.

**Weaknesses:**

- The paper appears to require daily generation of the dynamic road network topology using the tree-based adjacency matrix generation algorithm. The efficiency of this process remains unclear. Additionally, since the topologies undergo minimal changes between consecutive days, and substantial information is shared across these days, it raises the question of whether specialized algorithms are available to accelerate this topology generation.
- The authors present performance results only for two districts, D06 and D11. It is recommended to extend the reporting to include experimental results from the remaining seven districts.
- There is an inconsistency in the layout of the document: Figure 5 referred to on line 215 of Page 4, yet it is located on Page 7.
- The caption for Figure 7 is incorrect, and should be corrected to "Edge Dynamics" from "Node Dynamics".
- It is recommended that some recent related studies be discussed in the paper, particularly focusing on their performance with this dataset.

[1] UniST: A Prompt-Empowered Universal Model for Urban ST Prediction. KDD2024.
[2] Fine-Grained Urban Flow Prediction. WWW2021.
[3] When Transfer Learning Meets Cross-City Urban Flow Prediction: Spatio-Temporal Adaptation Matters. IJCAI2022.
[4] Spatio-Temporal Self-Supervised Learning for Traffic Flow Prediction. AAA2023.

**Questions:**

Please see the weaknesses for details. In addition,

- How does the topology generated by the tree-based method differ from that produced by the traditional method? It is recommended to provide statistical information to elucidate these differences.
- The rationale behind the design of parameter k in line 258 requires clarification. Further elucidation is needed regarding how this value affects the dataset generation.
- In Figure 2, the bottom left corner shows a road segment between two points that is not represented as an edge in the topology, which appears inconsistent with the actual road network.

---

### Official Review · Reviewer_3mAm · 2024-11-02

**Soundness:** 3
**Presentation:** 3
**Contribution:** 2
**Rating:** 5
**Confidence:** 4

**Summary:**

This paper introduces a novel dataset, DynST, aimed at addressing the challenges of traffic prediction in real-world scenarios where historical data is often limited. A key point of DynST is its evolving dynamic road network topology, which reflects real-world changes over time. This contrasts with traditional static datasets that use fixed network topologies, enhancing the dataset's relevance for practical applications.

**Strengths:**

1. This dataset is specifically for transfer learning in traffic prediction, addressing the gap in existing datasets which are often inadequate for such tasks.  The evolving nature of the road network topology represents an improvement from traditional static cases.

2. The dataset is extensive and spans two decades, providing a rich source of data for traffic forecasting research.

3. The authors conduct thorough experiments that not only demonstrate the utility of DynST but also validate their new adjacency matrix generation method.

**Weaknesses:**

1. The paper lacks a comprehensive comparison with existing state-of-the-art transfer learning approaches tailored for traffic forecasting.

2. While the authors argue for the benefits of evolving road networks, it is important to consider that the pace of road evolution is relatively slow. This raises questions about the necessity of transferring models over such long time scales. In many cases, retraining models on new road data may be more efficient and yield better performance than relying on transferred knowledge.

**Questions:**

See Weaknesses.

---

### Official Review · Reviewer_8GRo · 2024-11-04

**Soundness:** 2
**Presentation:** 2
**Contribution:** 1
**Rating:** 3
**Confidence:** 4

**Summary:**

This work presents a large-scale spatiotemporal dataset containing traffic data spanning 20 years across multiple regions. The primary objective of constructing this dataset is to provide sufficient source datasets for transfer learning.

**Strengths:**

(1) The dataset is massive in scale, encompassing traffic data from multiple regions across California over a 20-year period.

(2) The dataset contains information related to spatial dynamics, including node dynamics, edge dynamics, and graph lifecycle.

(3) A novel graph construction method is proposed, and experimental results demonstrate its superiority over traditional approaches in most scenarios.

**Weaknesses:**

(1) Compared to existing works (such as LargeST and STSGCN), this paper's main distinction lies in substantially increasing the volume of collected data. However, since all data is obtained through similar methods from the PEMS system, the contribution is somewhat limited.

(2) All data is sourced exclusively from California's PEMS system, without incorporating data from other regions or countries. More diverse data would be more meaningful for transfer learning tasks.

(3) While the research motivation is to provide rich source datasets for transfer learning in data-scarce target regions, this scenario is not adequately validated in the experimental phase. More realistic scenarios, such as training on California dataset (source dataset D06) and testing transfer learning on Chicago dataset (target dataset), would be more convincing.

(4) Although the paper considers the importance of dynamic road network topology, the dataset construction description and Figure 5 appear to reflect changes in sensors deployment rather than actual road network dynamics. Inferring road network dynamics based on sensors distribution does not align with real-world scenarios.

**Questions:**

Please refer to the Weaknesses.

---

### Official Review · Reviewer_BS5d · 2024-11-05

**Soundness:** 2
**Presentation:** 3
**Contribution:** 2
**Rating:** 5
**Confidence:** 4

**Summary:**

The authors introduce DynST, a comprehensive dataset comprising 20.35 billion data points collected over 20 years from 9 regions. It features an evolving dynamic road network topology that more accurately reflects real-world conditions. To enhance the representation of road networks, the paper introduces a novel tree-based algorithm for generating adjacency matrices. This new method overcomes the limitations of traditional distance-based approaches, which frequently result in either overconnected or disconnected nodes.

**Strengths:**

Existing datasets typically focus on non-transfer learning tasks and use fixed topological structures that do not accurately reflect the dynamic nature of real-world road networks. By proposing an evolving dynamic road network topology and a new tree-based algorithm for adjacency matrix generation, the authors creatively address the limitations of prior datasets.

The introduction of a tree-based algorithm improves upon traditional distance-based methods, which often result in inaccuracies.

The ability to transfer knowledge from data-rich regions to those with limited historical data can lead to better traffic predictions, potentially reducing congestion and improving transportation efficiency.

**Weaknesses:**

The proposed tree-based adjacency matrix generation method is introduced but lacks a thorough comparative evaluation against existing methods, particularly the distance-based method.

A deeper exploration of the underlying mechanics of this algorithm, including the rationale behind the choice of distance thresholds and how these thresholds impact connectivity, would provide valuable insight into its effectiveness. Additionally, discussing the computational complexity and efficiency of the proposed method compared to traditional distance-based approaches would help clarify its advantages.

Also, explaining how the adjacency matrix changes to reflect the dynamic nature of the road network—like how often it gets updated and the rules for adding or removing connections—would help readers better understand its real-world importance.

**Questions:**

Why does the proposed tree-based adjacency matrix generation method lack a thorough comparative evaluation against existing methods, particularly the distance-based method?

Could you provide a deeper exploration of the underlying mechanics of this algorithm? Specifically, what is the rationale behind the choice of distance thresholds, and how do these thresholds impact connectivity?

Can you discuss the computational complexity and efficiency of the proposed method compared to traditional distance-based approaches? What are the specific advantages of your method in this regard?

How does the adjacency matrix change to reflect the dynamic nature of the road network? How often does it get updated, and what rules govern the addition or removal of connections? How do these factors enhance its real-world applicability?

---

### Note · Authors · 2025-01-06

I have read and agree with the venue's withdrawal policy on behalf of myself and my co-authors.